# HistoPrism: Unlocking Functional Pathway Analysis from Pan-Cancer Histology via Gene Expression Prediction

**Susu Hu**[1,2,3,4,*] **Qinghe Zeng**[5,6]**, Nithya Bhasker**[1,2,3,4]**,**
**Jakob Nikolas Kather**[5,6,7,8]**, Stefanie Speidel**[1,2,3,4]

[1]Translational Surgical Oncology, National Center for Tumor Diseases (NCT/UCC) Dresden, Germany
[2]Faculty of Medicine and University Hospital Carl Gustav Carus, TUD Dresden University of Technology, Germany
[3]Helmholtz-Zentrum Dresden-Rossendorf (HZDR), Dresden, Germany
[4]German Cancer Research Center (DKFZ), Heidelberg, Germany
[5]Else Kroener Fresenius Center for Digital Health, Faculty of Medicine, TUD Dresden University of Technology, Germany
[6]Medical Oncology, National Center for Tumor Diseases (NCT), University Hospital Heidelberg, Heidelberg, Germany
[7]Department of Medicine I, Faculty of Medicine, TUD Dresden University of Technology, Germany
[8]Pathology & Data Analytics, Leeds Institute of Medical Research at St James's, University of Leeds, Leeds, United Kingdom

## Abstract

Predicting spatial gene expression from H&E histology offers a scalable and clinically accessible alternative to sequencing, but realizing clinical impact requires models that generalize across cancer types and capture biologically coherent signals. Prior work is often limited to per-cancer settings and variance-based evaluation, leaving functional relevance underexplored. We introduce HistoPrism, an efficient transformer-based architecture for pan-cancer prediction of gene expression from histology. To evaluate biological meaning, we introduce a pathway-level benchmark, shifting assessment from isolated gene-level variance to coherent functional pathways. HistoPrism not only surpasses prior state-of-the-art models on highly variable genes , but also more importantly, achieves substantial gains on pathway-level prediction, demonstrating its ability to recover biologically coherent transcriptomic patterns. With strong pan-cancer generalization and improved efficiency, HistoPrism establishes a new standard for clinically relevant transcriptomic modeling from routinely available histology. *Code is available at https://github.com/susuhu/HistoPrism.*

## 1 Introduction

Spatial transcriptomics (ST) combines high-resolution imaging with transcriptomic profiling to map the spatial distribution of gene expression within intact tissues (Khan et al., 2024). By preserving spatial context, ST has enabled advances across developmental biology, oncology, immunology, and histopathology (Choe et al., 2023). However, ST remains costly, labor-intensive, and not yet widely scalable. In contrast, hematoxylin and eosin (H&E) stained whole-slide images (WSIs) are routinely acquired in clinical workflows, motivating computational approaches to infer spatial gene expression directly from histology for cost-effective and scalable histogenomic analysis.

Early approaches to this problem often relied on complex, multi-stage pipelines involving brittle learning heuristics such as contrastive learning with ill-defined negative samples (Xie et al., 2023; Long et al., 2023), retrieval-based inference schemes that limit generalization (Xie et al., 2023), or intricate multi-resolution engineering with significant computational overhead (Chung et al., 2024). Generative and contextual approaches, including diffusion-based STEM (Zhu et al., 2025) and flow-based STFlow (Huang et al., 2025a), model the uncertainty of one-to-many mapping between WSIs and gene expressions, but have been limited to single-cancer settings and are computationally intensive. Pan-cancer models, such as STPath (Huang et al., 2025b), achieve zero-shot generalization using masked gene prediction on large-scale datasets. Nevertheless, they rely on stable gene-gene correlations, which can be inconsistent across heterogeneous tissues and sequencing techniques.

---

*Correspondence to Susu Hu: `susu.hu@nct-dresden.de`.

Moreover, the evaluation of predicted gene expression has largely focused on Pearson correlation of top-N highly variable genes, neglecting functional coherence.

To address these gaps, we introduce HistoPrism, an efficient transformer-based architecture for pan-cancer gene expression prediction, alongside Gene Pathway Coherence (GPC), a new evaluation framework based on 50 Hallmark gene sets and 87 Gene Ontology pathway gene sets. GPC quantifies the biological fidelity of predictions by assessing pathway-level coherence, moving beyond variance-based metrics. Our pan-cancer benchmark shows that HistoPrism delivers state-of-the-art performance in both top-N variable gene prediction and pathway-focused prediction, while maintaining a substantially smaller and more computationally efficient footprint. Crucially, pathway-focused evaluation is key for identifying models suitable for clinical use, as it prioritizes biological interpretability rather then relying solely on aggregated accuracy.

## 2 RELATED WORK

### 2.1 COMPUTATIONAL PREDICTION OF SPATIAL TRANSCRIPTOMICS

**Regression-Based Approaches.** Early methods typically framed histology-to-gene prediction as a regression problem. BLEEP (Xie et al., 2023) employs contrastive learning to align paired histology and gene expression into a joint embedding space, enabling inference via nearest-neighbor matching. However, defining negative pairs in pathology remains ambiguous, and retrieving-based inference limits generalization to unseen queries. GraphST (Long et al., 2023) incorporates spatial structure through graph neural networks (GNNs), but inherits similar weaknesses from contrastive training. TRIPLEX (Chung et al., 2024) introduces a multi-resolution architecture with distillation losses to capture both local and global context, yet its complexity results in high computational cost and reduced interpretability.

**Generative Approaches.** More recent work has reframed this task through generative modeling (Zhu et al., 2025; Huang et al., 2025a), motivated by the inherently one-to-many mapping from histology to gene expression (Zhu et al., 2025). These methods aim to capture distributions of plausible expression profiles, but have thus far been validated primarily in single-cancer settings. Extending them to pan-cancer prediction introduces a far greater challenge, where heterogeneity across multiple cancer types raises concerns of scalability and mode collapse.

A notable advance is STPath (Huang et al., 2025b), a pan-cancer foundation model built on a BERT-style framework (Devlin et al., 2019). Using masked-gene modeling on a massive 38k gene panel, STPath learns complex contextual dependencies and achieved a new state-of-the-art on standard variance-based benchmarks. However, this strategy implicitly assumes that gene–gene correlations are stable signals, an assumption that often breaks down in heterogeneous, tissue-specific pan-cancer settings. In practice, the model's considerable size makes training and fine-tuning highly resource-intensive, which can be a barrier for broad adoption and adaptation in many research and clinical settings.

**Our Approach.** We propose HistoPrism, a transformer-based model that leverages rich visual features to predict gene expression in pan-cancer datasets. Its design effectively captures visual–molecular relationships and supports pathway-level prediction coherence, achieving state-of-the-art predictive performance while being more efficient and practical for clinical deployment than previous approaches.

### 2.2 FOUNDATION MODELS IN DIGITAL PATHOLOGY

The advent of self-supervised learning on massive, gigapixel-scale datasets has given rise to powerful Pathology Foundation Models (PFMs). Models such as CTransPath (Wang et al., 2022), GigaPath (Xu et al., 2024), and UNI (Chen et al., 2024b) are pre-trained on millions of histology patches, learning rich visual representations of tissue morphology that are highly effective for a wide range of downstream tasks. These PFMs serve as a crucial backbone for modern computational pathology, including the prediction of spatial gene expression. With PFMs standardizing the extraction of high-quality, patch-level visual features, the core research challenge shifts from feature engineering to the subsequent problem: modeling how these patch representations can be contextually integrated

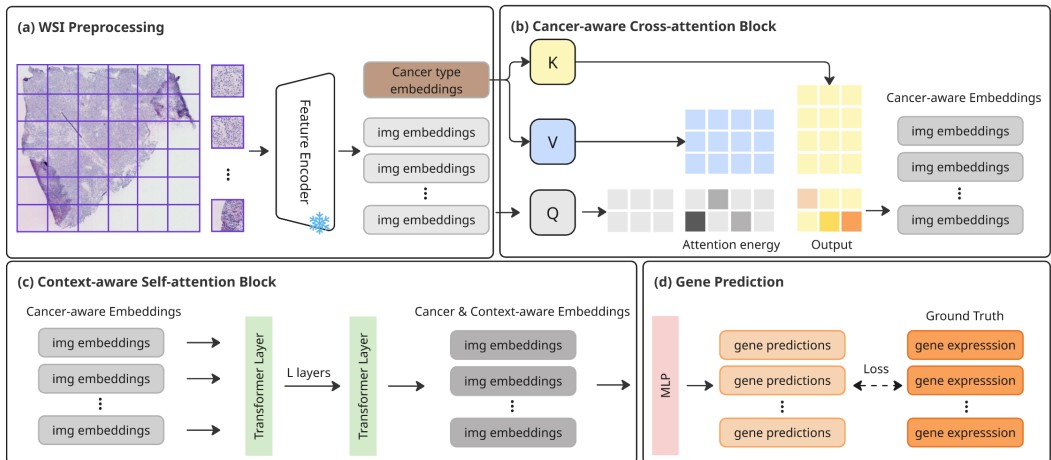

Figure 1: **HistoPrism architecture**. Patch level image embeddings are obtained via pathology foundation models. A cross-attention module injects pan-cancer conditioning. A Transformer Encoder models contextual relations before a final MLP head regresses gene expression values.

and spatially structured to capture the underlying biology of the tumor microenvironment. Our work directly addresses this challenge.

# 3 METHODOLOGY

In this section, we first formally define the problem, then detail the HistoPrism architecture, its training objective and our gene pathway coherence evaluation framework.

## 3.1 PROBLEM FORMULATION

We consider an H&E-stained whole-slide image, divided into $N$ non-overlapping patches. Each patch is represented by a feature vector $\mathbf{x}_i \in \mathbb{R}^{D_{img}}$, extracted by a pre-trained pathology foundation model (PFM). Spatial transcriptomics provides the corresponding raw count vector of gene expressions, which we normalize as $\mathbf{y}_i \in \mathbb{R}^{D_{gene}}$ using a log1p transformation. Additionally, each slide is associated with a global condition cancer type, encoded as a one-hot vector $\mathbf{c} \in \{0,1\}^{D_{onco}}$.

The goal is to learn a parameterized mapping function $f_\theta : (\mathbb{R}^{N \times D_{img}}, \mathbb{R}^{D_{onco}}) \rightarrow \mathbb{R}^{N \times D_{gene}}$ that predicts gene expression from H&E image features. For each input patch feature $\mathbf{x}_i$, the model outputs gene expression vector $\hat{\mathbf{y}}_i = f_\theta(\mathbf{X}, \mathbf{c})_i$, where $\mathbf{X}$ denotes the set of all patch embeddings. The model parameters $\theta$ are optimized to minimize the difference between the predicted expression $\hat{\mathbf{Y}} = \{\hat{\mathbf{y}}_1, \ldots, \hat{\mathbf{y}}_N\}$ and the ground-truth expression $\mathbf{Y} = \{\mathbf{y}_1, \ldots, \mathbf{y}_N\}$.

## 3.2 HISTOPRISM: A DIRECT-MAPPING ARCHITECTURE

HistoPrism is a transformer-based regressor designed for efficient and direct mapping from visual features to gene expression. It eschews the complex contextual reconstruction of prior work in favor of a streamlined architecture that models cancer-aware contextualized pathology image features for corresponding gene profiles. The architecture, depicted in Figure 1, consists of three main stages.

**1. Pan-Cancer Conditioning via Cross-Attention.** To make the model aware of the global cancer type, we condition the visual features using a cross-attention mechanism. The one-hot cancer type vector $\mathbf{c}$ is first projected into a dense embedding $\mathbf{c}_{\text{emb}} \in \mathbb{R}^{D_{img}}$ via a linear layer. This embedding serves as the context for the cross-attention module, providing the Key ($\mathbf{K}$) and Value ($\mathbf{V}$), while

the patch features $\mathbf{X}$ serve as the Query ($\mathbf{Q}$).

$$\mathbf{Q} = \mathbf{X}\mathbf{W}_Q \tag{1}$$

$$\mathbf{K}, \mathbf{V} = \mathbf{c}_{\text{emb}}\mathbf{W}_K, \quad \mathbf{c}_{\text{emb}}\mathbf{W}_V \tag{2}$$

$$\mathbf{X}_{\text{cond}} = \text{CrossAttention}(\mathbf{Q}, \mathbf{K}, \mathbf{V}) \tag{3}$$

This allows the model to modulate the patch representations based on the overarching cancer type, enabling it to learn both pan-cancer and cancer-specific histopathological patterns.

**2. Contextual Aggregation with a Transformer Encoder.** The conditioned patch features $\mathbf{X}_{\text{cond}}$ are first projected into a hidden dimension $D_{hidden}$ and then processed by a standard Transformer Encoder (Vaswani et al., 2017). This module captures both short and long-range spatial dependencies between patches, modeling higher-level tissue structures such as tumor boundaries and immune infiltration patterns. The output of the transformer, $\mathbf{H}_{\text{latent}} \in \mathbb{R}^{N \times D_{hidden}}$, is a set of contextually rich latent representations for each patch.

**3. Gene Expression Regression.** Finally, a multi-layer perceptron (MLP) serves as the regression head. It takes the latent representation $\mathbf{h}_i \in \mathbf{H}_{\text{latent}}$ for each patch and maps it directly to the predicted $D_{gene}$-dimensional gene expression vector $\hat{\mathbf{y}}_i$.

$$\hat{\mathbf{y}}_i = \text{MLP}_{\text{head}}(\mathbf{h}_i) \tag{4}$$

HistoPrism is trained end-to-end by minimizing the Mean Squared Error (MSE) $\mathcal{L}_{\text{MSE}}$ between the predicted and ground-truth gene expression values across all $N$ patches.

$$\mathcal{L}_{\text{MSE}} = \frac{1}{N} \sum_{i \in N} (\hat{y}_i - y_i)^2 \tag{5}$$

Our design favors direct feature fusion over contrastive alignment for regression tasks, employing self-attention to robustly aggregate sparse biological signals from variable-sized tissue patches where standard pooling fails.

### 3.3 A FRAMEWORK FOR EVALUATING BIOLOGICAL COHERENCE

To rigorously assess model performance, we employ a two-tiered evaluation strategy. We first use the standard metric for comparability with prior work and then introduce our proposed benchmark, which is designed to measure a model's ability to predict biologically meaningful expression patterns.

**Baseline Metric: Highly Variant Gene Correlation.** The standard protocol in this domain is to evaluate the Pearson Correlation Coefficient (PCC) between predicted and ground-truth expression for the top-$N$ most highly-variant genes (HVGs) across a test set. While this metric is useful for gauging a model's ability to capture high-magnitude signals, its clinical and biological relevance is limited. It focuses on a small, statistically-driven subset of genes, ignoring thousands of others, and it fails to measure whether a model has learned the *coordinated* expression patterns that define a functional biological process. A model can achieve a high HVG while failing to generate biologically coherent predictions, thus limiting its translational potential.

**The Gene Pathway Coherence (GPC) Benchmark.** To address the limitations of variance-based metrics, we propose the Gene Pathway Coherence (GPC) benchmark. Our goal is to bridge the gap between standard machine learning evaluation and the principles of biological inquiry. While computational biology has long relied on pathway analysis to understand function, this approach has not yet been formalized as a standard benchmark for deep learning models in this domain. The GPC benchmark is designed to fill this void. It assesses a model's ability to reconstruct the coordinated expression of functionally related genes, thereby aligning the evaluation protocol with the true scientific objective of understanding cellular function.

The construction of our benchmark follows a rigorous, multi-stage curation process:

1. **Source Curation:** We begin by aggregating a comprehensive set of pathways from two authoritative, widely-used biological databases: the Hallmark gene sets from the Molecular Signatures Database (MSigDB) (Broad Institute, 2025), which represent well-defined biological states or processes, and the Gene Ontology (GO) database (Gene Ontology Consortium, 2025), from which we include terms for Biological Process (BP), Cellular Component (CC), and Molecular Function (MF).

2. **Size Filtering:** Recognizing that these collections contain thousands of pathways of varying size, we first filter for those of a tractable and meaningful length. We retain only pathways containing between 50 and 100 genes, a range that avoids both overly specific sets prone to noise and overly broad pathways. Hallmark pathways are retained in full, as there are only 50.

3. **Redundancy Filtering:** To create a non-redundant benchmark, we address the significant topical overlap between pathways. We compute the Jaccard similarity, $J(A, B) = |A \cap B|/|A \cup B|$, for all pairs of pathways $(A, B)$ based on their member genes. For any pair where the similarity exceeds a threshold of $\tau = 0.1$, we iteratively remove the larger of the two pathways until no pairs violate this condition.

For each pathway, the evaluation score is the computed across all of its member genes. Let $\mathcal{Y} = \{(\mathbf{y}_i, \hat{\mathbf{y}}_i)\}_{i=1}^{N}$ denote the paired ground-truth and predicted gene expression sets for $N$ whole-slide images (WSIs). Each WSI $i$ contains $n_i$ patches, with $\mathbf{y}_i = [\mathbf{y}_{i1}, \ldots, \mathbf{y}_{in_i}]^{\top}$ and $\hat{\mathbf{y}}_i = [\hat{\mathbf{y}}_{i1}, \ldots, \hat{\mathbf{y}}_{in_i}]^{\top}$, where $\mathbf{y}_{ij}, \hat{\mathbf{y}}_{ij} \in \mathbb{R}^{D_{\text{gene}}}$ represent the expression vectors of $D_{\text{gene}}$ genes.

For each gene $g \in \{1, \ldots, D_{\text{gene}}\}$ within WSI $i$, we compute the Pearson correlation coefficient (PCC) across all patches:

$$r_{i,g} = \frac{\text{cov}(\mathbf{y}_{i,:,g}, \hat{\mathbf{y}}_{i,:,g})}{\sigma(\mathbf{y}_{i,:,g}) \, \sigma(\hat{\mathbf{y}}_{i,:,g})}, \quad (6)$$

where $\mathbf{y}_{i,:,g} = [y_{i1g}, \ldots, y_{in_ig}]^{\top}$ and $\hat{\mathbf{y}}_{i,:,g} = [\hat{y}_{i1g}, \ldots, \hat{y}_{in_ig}]^{\top}$ denote the expression profiles of gene $g$ across all patches of WSI $i$.

Given a curated collection of $M$ gene pathways $\mathcal{P} = \{P_1, \ldots, P_M\}$, where each $P_m \subseteq \{1, \ldots, D_{\text{gene}}\}$ indexes the genes in pathway $m$, the final pathway-level coherence score is defined as

$$s_m = \frac{1}{N} \sum_{i=1}^{N} \frac{1}{|P_m|} \sum_{g \in P_m} r_{i,g}. \quad (7)$$

By evaluating with biologically coherent patterns rather than variance alone, this framework yields a clinically relevant perspective on evaluating ST prediction performance.

## 4 EXPERIMENTS AND RESULTS

### 4.1 EXPERIMENTAL SETUP

We conduct experiments on the HEST1k dataset (Jaume et al., 2024), using two splits that retain the original hold-out test splits from HESKT1k HEST-Bench. Training and validation splits are stratified by cancer type. HEST1k is a large-scale dataset aggregating 153 distinct cohorts from 36 independent studies. This collection encapsulates high inter-center variability, including diverse spatial transcriptomics technologies, staining protocols, and scanner vendors, ensuring that the holdout evaluation reflects true cross-center generalization. We also considered STimage-1K4M (Chen et al., 2024a), another large-scale resource. However, we determined it was unsuitable for this study due to its use of non-standard, single-resolution image formats and its partial data overlap with HEST1k.

STPath serves as our primary benchmark, as it outperforms MLP with UNI and GigaPath PFM, as well as two deep learning methods BLEEP, and TRIPLEX in pan-cancer gene prediction. Due to limited computational resources and the unavailability of the STPath training code, we only performed inference using their corresponding PFM GigaPath (Xu et al., 2024), which aligns with STPath's intended use as a foundation model for inference.

Since state-of-the-art regression models have already been extensively evaluated in STPath, we focus on extending our comparison with recent generative approaches. Specifically, we include STEM

(Zhu et al., 2025), a diffusion-based model, and STFlow(Huang et al., 2025a), a flow-matching generative model. Both were originally benchmarked on single-cancer datasets, whereas we evaluate their generalization in a more challenging pan-cancer setting. Due to the computational cost of STEM and STFlow training, we restrict both models to the union of the top 50 highly variable genes across all cancer types. Although this smaller gene subset emphasizes the most variable signals, STEM performs significantly worse than other methods, calling into question the robustness of its original leave-one-out evaluation. Similarly, STFlow struggles to generalize beyond single-cancer settings, underscoring the limitations of current generative models in capturing complex multimodal relationships between histology and gene expression across diverse tumor types.

Our proposed model HistoPrism consists of 1 cross attention layer with 4 heads and 2 transformer layers with 8 heads and 256 hidden dimension receptively. HistoPrism is trained end-to-end with UNI PFM (Chen et al., 2024b) with a gene panel of size 38,982 curated by STPath. The training details are included in Appendix B.

## 4.2 HISTOPRISM ACHIEVES STATE-OF-THE-ART PAN-CANCER PERFORMANCE

We first evaluate pan-cancer gene prediction performance on the top 50 highly variable genes (HVGs) using Pearson correlation coefficient (PCC). As expected, STEM performs poorly in the pan-cancer setting, likely because diffusion-based models struggle to capture the high heterogeneity and complex multi-modal relationships between histology and gene expression across diverse cancer types. In Table 1, we report both macro-average PCC, computed as the mean PCC across the two splits for each cancer type, and micro-average PCC, computed across all individual samples to account for class imbalance. Macro-average treats each cancer type equally, while micro-average reflects overall predictive performance weighted by sample counts. Detailed sample counts can be found in the Appendix 6. HistoPrism demonstrates competitive performance, slightly below STPath on macro-average PCC but higher on micro-average PCC. Since micro-average PCC captures performance across all samples, it provides a more balanced view of predictive quality, highlighting HistoPrism's robustness across heterogeneous cancers.

Table 1: Macro- and Micro-Average PCC ↑ of Top50 HVGs across 10 different cancer types. Best in **bold**.

| Cancer Type | Macro-Average PCC | | | | Micro-Average PCC | | | |
|---|---|---|---|---|---|---|---|---|
| | STPath | STFlow[*] | STEM[*] | HistoPrism | STPath | STFlow[*] | STEM[*] | HistoPrism |
| CCRCC | $0.117_{0.001}$ | $0.140_{0.002}$ | $0.124_{0.029}$ | $\mathbf{0.206}_{0.008}$ | $0.117_{0.146}$ | $0.140_{0.126}$ | $0.123_{0.041}$ | $\mathbf{0.206}_{0.102}$ |
| COAD | $\mathbf{0.393}_{0.185}$ | $0.346_{0.135}$ | $0.236_{0.001}$ | $0.353_{0.125}$ | $\mathbf{0.459}_{0.132}$ | $0.394_{0.098}$ | $0.235_{0.021}$ | $0.397_{0.096}$ |
| HCC | $0.094_{0.052}$ | $0.070_{0.001}$ | $0.098_{0.032}$ | $\mathbf{0.113}_{0.014}$ | $0.094_{0.052}$ | $0.070_{0.001}$ | $0.098_{0.032}$ | $\mathbf{0.113}_{0.014}$ |
| IDC | $\mathbf{0.629}_{0.126}$ | $0.547_{0.080}$ | $0.178_{0.029}$ | $0.477_{0.019}$ | $\mathbf{0.629}_{0.126}$ | $0.547_{0.080}$ | $0.178_{0.029}$ | $0.477_{0.019}$ |
| LUNG | $\mathbf{0.518}_{0.028}$ | $0.468_{0.075}$ | $0.220_{0.018}$ | $0.498_{0.020}$ | $\mathbf{0.518}_{0.028}$ | $0.468_{0.075}$ | $0.220_{0.018}$ | $0.498_{0.020}$ |
| LYMPH_IDC | $0.182_{0.075}$ | $0.185_{0.034}$ | $0.160_{0.010}$ | $\mathbf{0.215}_{0.042}$ | $0.182_{0.075}$ | $0.185_{0.034}$ | $0.160_{0.010}$ | $\mathbf{0.215}_{0.042}$ |
| PAAD | $\mathbf{0.493}_{0.100}$ | $0.420_{0.203}$ | $0.195_{0.064}$ | $0.420_{0.019}$ | $\mathbf{0.493}_{0.100}$ | $0.420_{0.203}$ | $0.195_{0.064}$ | $0.420_{0.019}$ |
| PRAD | $0.257_{0.012}$ | $0.202_{0.010}$ | $0.185_{0.056}$ | $\mathbf{0.324}_{0.030}$ | $0.255_{0.139}$ | $0.200_{0.107}$ | $0.184_{0.070}$ | $\mathbf{0.317}_{0.134}$ |
| READ | $0.280_{0.030}$ | $0.228_{0.075}$ | $0.218_{0.012}$ | $\mathbf{0.295}_{0.023}$ | $0.279_{0.024}$ | $0.228_{0.071}$ | $0.220_{0.029}$ | $\mathbf{0.295}_{0.037}$ |
| SKCM | $\mathbf{0.588}_{0.113}$ | $0.503_{0.102}$ | $0.228_{0.068}$ | $0.523_{0.046}$ | $\mathbf{0.588}_{0.113}$ | $0.503_{0.102}$ | $0.228_{0.068}$ | $0.523_{0.046}$ |
| Average | $\mathbf{0.361}_{0.072}$ | $0.311_{0.167}$ | $0.184_{0.004}$ | $0.342_{0.138}$ | $0.292_{0.191}$ | $0.247_{0.157}$ | $0.180_{0.064}$ | $\mathbf{0.318}_{0.138}$ |

[*] STEM and STFlow are trained only with 430 union top50 HVG genes due to limited computing resources.

## 4.3 BEYOND VARIANCE: HISTOPRISM CAPTURES COHERENT BIOLOGY IN LOW-VARIANCE PATHWAYS

We evaluated gene pathway coherence (GPC) for HistoPrism and STPath, on both Hallmark gene pathways and Gene Ontology pathways. HistoPrism demonstrates consistent gains, outperforming STPath on 86.0% of the 50 Hallmark pathways and on 74.7% of the Gene Ontology pathways. Beyond these overall win rates, stratifying pathways by variance level (Figure 2) reveals a more fundamental distinction. HistoPrism achieves its largest gains on low-variance pathways, which are often associated with stable, core biological processes(Eisenberg & Levanon, 2013).

This comparison underscores a fundamental difference in modeling strategy: while STPath primarily leverages the most variable signals, HistoPrism's direct-mapping architecture effectively captures both high-variance genes and the subtler, coordinated expression patterns that define cellular programs. These findings suggest that isolated gene-level variance-based metrics provide an incomplete

assessment of a model's ability to reconstruct biologically meaningful gene expression. More details can be found in Appendix C.

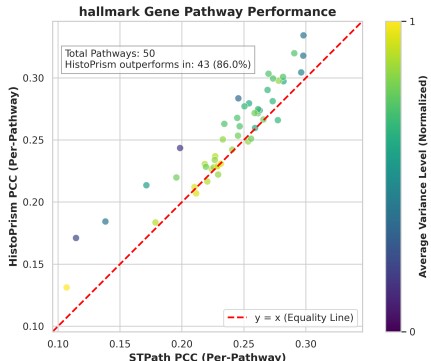

(a) Hallmark gene pathway performance.

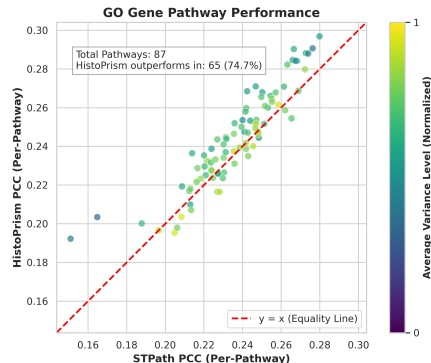

(b) Gene ontology pathway performance.

Figure 2: Comparison of gene pathway coherence (GPC) in PCC on both Hallmark gene pathways and Gene Ontology pathways.

## 4.4 HOLISTIC ASSESSMENT OF PREDICTED GENE EXPRESSION

We further evaluate the biological relevance of the predicted expression profiles by clustering all samples based on their predicted expression across the full set of 38k genes, and comparing the resulting clusters to the ground-truth cancer type labels. Table 2 reports Adjusted Mutual Information (AMI) and Adjusted Rand Index (ARI) between the predicted clusters and the true cancer types. Due to class imbalance, AMI serves as the more informative measure, while ARI is reported for completeness. This evaluation provides a holistic view of prediction quality beyond subset-based assessments, as successful clustering requires the model to generate a biologically coherent representation across the entire transcriptome. HistoPrism achieves substantially higher scores than STPath, which we attribute to its architectural design. By contrast, the "fill-in-the-blanks" objective of STPath, based on a masked autoencoder, is architecturally optimized for reconstruction and imputation. For a pure predictive task where no gene information is provided at inference, this framework is suboptimal. Our proposed direct mapping is more naturally suited for this modality translation task, avoiding the inductive bias of an autoencoder on a generative problem.

Table 2: Quantitative comparison of downstream clustering utility (AMI/ARI). Best in **bold**.

| Model | AMI $\uparrow$ | ARI $\uparrow$ |
|---|---|---|
| STPath | $0.395_{0.523}$ | $0.402_{0.016}$ |
| HistoPrism | $\mathbf{0.623}_{0.015}$ | $\mathbf{0.521}_{0.001}$ |

## 4.5 DATA-EFFICIENCY AND SCALABILITY ANALYSIS

To assess computational efficiency, we benchmarked HistoPrism against the baseline STPath across forward-pass runtime, peak GPU memory, and FLOPs (Figure. 3). Both models used identical image and gene embedding dimension and the same number of patches. Profiling shows that HistoPrism consistently requires fewer FLOPs, less memory, and shorter runtimes than STPath, with the gap widening as patch counts increase. Notably, HistoPrism scales linearly across all three metrics, while STPath exhibits exponential growth, highlighting HistoPrism's deployment efficiency for real-world datasets exceeding 10k patches. Crucially, HistoPrism achieves this performance while being trained on only 500 whole-slide images, roughly half the data used for the STPath foundation model, underscoring its remarkable data efficiency. These efficiency gains are especially critical in clinical settings, where computational resources are often limited, making HistoPrism a practical and scalable solution for whole-slide image analysis. All experiments were run on a single NVIDIA A100

GPU with 100-run averages. FLOPs and peak memory show no variance, and the inference-time standard deviation is negligible.

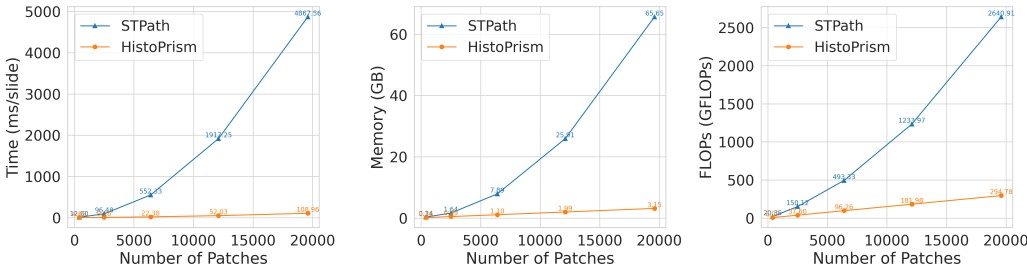

Figure 3: Model efficiency comparison of HistoPrism and STPath in terms of forward pass runtime, peak GPU memory usage, and FLOPs across different numbers of patches.

## 4.6 ABLATION STUDY

We conducted an ablation study to disentangle the contributions of cross-attention and explicit spatial priors. As shown in Table 3, conditioning on cancer type through cross-attention consistently improves performance, highlighting the importance of modulating local representations with global context. Surprisingly, however, adding explicit positional encoding (PE) yields no measurable benefit, contrary to common assumptions in Transformer-based architectures. We hypothesize two reasons for this. First, the prediction task is predominantly local: the rich latent features extracted by UNI PFM already capture morphology within and around each patch, leaving little additional signal to be gained from absolute spatial coordinates. Second, in the absence of PE, the Transformer behaves as a permutation-invariant set function, effectively leveraging the global compositional structure of the tissue without being anchored to fixed positions.

Table 3: Ablation study of cross attention and positional encoding with predictive accuracy in PCC ↑ on top50 HVGs. Best in **bold**.

| Cancer Type | Macro-Average PCC | | | Micro-Average PCC | | |
|---|---|---|---|---|---|---|
| | w/o CrossAttn | HistoPrism+PE | HistoPrism | w/o CrossAttn | HistoPrism+PE | HistoPrism |
| CCRCC | $0.227_{0.039}$ | $\mathbf{0.236}_{0.047}$ | $0.206_{0.008}$ | $\mathbf{0.498}_{0.036}$ | $0.236_{0.133}$ | $0.206_{0.102}$ |
| COAD | $0.289_{0.026}$ | $\mathbf{0.385}_{0.118}$ | $0.353_{0.125}$ | $0.299_{0.047}$ | $\mathbf{0.427}_{0.088}$ | $0.397_{0.096}$ |
| HCC | $0.102_{0.004}$ | $0.108_{0.028}$ | $\mathbf{0.113}_{0.014}$ | $0.102_{0.004}$ | $0.108_{0.028}$ | $\mathbf{0.113}_{0.014}$ |
| IDC | $0.443_{0.030}$ | $0.420_{0.053}$ | $\mathbf{0.477}_{0.019}$ | $0.443_{0.030}$ | $0.420_{0.053}$ | $\mathbf{0.477}_{0.019}$ |
| LUNG | $0.465_{0.028}$ | $0.482_{0.022}$ | $\mathbf{0.498}_{0.020}$ | $0.465_{0.028}$ | $0.482_{0.022}$ | $\mathbf{0.498}_{0.020}$ |
| LYMPH_IDC | $\mathbf{0.230}_{0.067}$ | $0.208_{0.051}$ | $0.215_{0.042}$ | $\mathbf{0.230}_{0.067}$ | $0.208_{0.051}$ | $0.215_{0.042}$ |
| PAAD | $0.368_{0.049}$ | $0.394_{0.033}$ | $\mathbf{0.420}_{0.019}$ | $0.368_{0.049}$ | $0.394_{0.033}$ | $\mathbf{0.420}_{0.019}$ |
| PRAD | $\mathbf{0.334}_{0.038}$ | $0.321_{0.054}$ | $0.324_{0.030}$ | $\mathbf{0.325}_{0.134}$ | $0.310_{0.134}$ | $0.317_{0.134}$ |
| READ | $0.224_{0.121}$ | $0.258_{0.008}$ | $\mathbf{0.295}_{0.023}$ | $0.224_{0.104}$ | $0.258_{0.039}$ | $\mathbf{0.295}_{0.037}$ |
| SKCM | $0.498_{0.036}$ | $0.494_{0.040}$ | $\mathbf{0.523}_{0.046}$ | $0.498_{0.036}$ | $0.494_{0.040}$ | $\mathbf{0.523}_{0.046}$ |
| **Average** | $0.318_{0.129}$ | $0.331_{0.129}$ | $\mathbf{0.342}_{0.138}$ | $0.306_{0.134}$ | $0.313_{0.137}$ | $\mathbf{0.318}_{0.138}$ |

To ensure a fair comparison with STPath, we further ablated our model by replacing our PFM with the Gigapath as used in STPath. As shown in Table 4 and Figure4, this substitution results in only marginal performance differences, indicating that our approach does not rely heavily on pretrained PFM representations. Therefore, we exclude the use of Gigapath in the main experiments to isolate the contribution of our architecture rather than external foundation model priors.

## 5 DISCUSSION

We introduced HistoPrism, a direct-mapping transformer for pan-cancer spatial transcriptomics prediction, together with Gene Pathway Coherence (GPC), a benchmark that aligns evaluation with biological function. Variance-based metrics, while useful, are a poor proxy for coordinated cellu-

Table 4: Ablation study of PFMs and positional encoding with predictive accuracy in PCC ↑ on top50 HVGs. Best in **bold**.

| Cancer Type | Macro-Average PCC | | | Micro-Average PCC | | |
|---|---|---|---|---|---|---|
| | **STPath** | **HistoPrism+GigaPath** | **HistoPrism** | **STPath** | **HistoPrism+GigaPath** | **HistoPrism** |
| CCRCC | $0.117_{0.001}$ | $\mathbf{0.214}_{0.023}$ | $0.206_{0.008}$ | $0.117_{0.146}$ | $\mathbf{0.214}_{0.127}$ | $0.206_{0.102}$ |
| COAD | $\mathbf{0.393}_{0.185}$ | $0.331_{0.123}$ | $0.353_{0.125}$ | $\mathbf{0.459}_{0.132}$ | $0.375_{0.098}$ | $0.397_{0.096}$ |
| HCC | $0.094_{0.052}$ | $0.096_{0.015}$ | $\mathbf{0.113}_{0.014}$ | $0.094_{0.052}$ | $0.096_{0.015}$ | $\mathbf{0.113}_{0.014}$ |
| IDC | $\mathbf{0.629}_{0.126}$ | $0.472_{0.037}$ | $0.477_{0.019}$ | $\mathbf{0.629}_{0.126}$ | $0.472_{0.037}$ | $0.477_{0.019}$ |
| LUNG | $\mathbf{0.518}_{0.028}$ | $0.502_{0.015}$ | $0.498_{0.020}$ | $\mathbf{0.518}_{0.028}$ | $0.502_{0.015}$ | $0.498_{0.020}$ |
| LYMPH_IDC | $0.182_{0.075}$ | $\mathbf{0.227}_{0.060}$ | $0.215_{0.042}$ | $0.182_{0.075}$ | $\mathbf{0.227}_{0.060}$ | $0.215_{0.042}$ |
| PAAD | $\mathbf{0.493}_{0.100}$ | $0.401_{0.008}$ | $0.420_{0.019}$ | $\mathbf{0.493}_{0.100}$ | $0.401_{0.008}$ | $0.420_{0.019}$ |
| PRAD | $0.257_{0.012}$ | $\mathbf{0.346}_{0.048}$ | $0.324_{0.030}$ | $0.255_{0.139}$ | $\mathbf{0.336}_{0.138}$ | $0.317_{0.134}$ |
| READ | $0.280_{0.030}$ | $0.263_{0.055}$ | $\mathbf{0.295}_{0.023}$ | $0.279_{0.028}$ | $0.263_{0.057}$ | $\mathbf{0.295}_{0.037}$ |
| SKCM | $\mathbf{0.588}_{0.113}$ | $0.460_{0.169}$ | $0.523_{0.046}$ | $\mathbf{0.588}_{0.113}$ | $0.460_{0.169}$ | $0.523_{0.046}$ |
| **Average** | $\mathbf{0.361}_{0.072}$ | $0.331_{0.139}$ | $0.342_{0.138}$ | $0.292_{0.191}$ | $\mathbf{0.320}_{0.142}$ | $0.318_{0.138}$ |

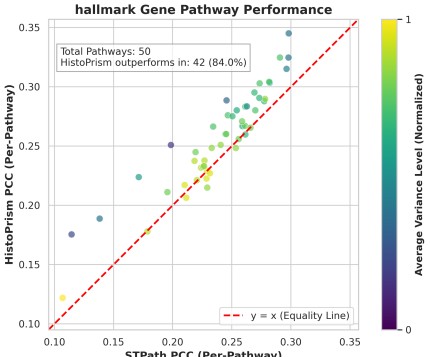

(a) Hallmark gene pathway performance.

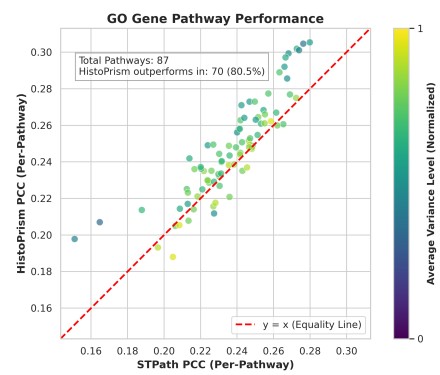

(b) Gene ontology pathway performance.

Figure 4: Ablation study of the impact of PFM Gigapath on our model HistoPrism's GPC performance.

lar processes. By shifting to pathway-level structure, GPC provides a more rigorous measure of performance.

Across experiments, HistoPrism outperforms strong baselines, including STPath, STFlow and STEM, not only on highly variable genes but also at the pathway level, where biological coherence is critical. Global evaluation using 38,928 gene clustering further shows large gains in AMI and ARI, demonstrating that HistoPrism captures both fine-grained gene programs and broad cancer-type organization. Crucially, we demonstrate that these gains are architectural and independent of the underlying feature extractor: while we utilized UNI features for benchmarking consistency, our ablations confirm that HistoPrism remains robust and effective when trained with GigaPath.

Beyond predictive performance, HistoPrism is optimized for resource-constrained settings, achieving SOTA performance with only approximately 50% of standard training data and a minimal computational footprint. This efficiency directly supports clinical deployment in institutes where large-scale compute or massive annotated datasets are unavailable.

While our work establishes a robust predictive model, a key avenue for future research is to enhance its biological interpretability. Moving beyond predictive accuracy to systematically identify the causal visual features and cellular concepts the model has learned will be crucial for its adoption as a tool for scientific discovery.

# 6 CONCLUSION

We presented HistoPrism, an efficient transformer for pan-cancer prediction of gene expression from histology. HistoPrism achieves state-of-the-art accuracy on highly variable genes, stronger biolog-

ical coherence at the pathway level, and superior global clustering performance (AMI, ARI) across 38k genes. In addition to accuracy and fidelity, HistoPrism offers major efficiency gains, enabling large-scale pan-cancer analysis at lower cost. By introducing GPC, we move evaluation beyond variance-based metrics toward functional interpretability, a prerequisite for clinical relevance. Together, these advances highlight HistoPrism's potential to bridge histology and transcriptomics at scale, bringing computational spatial genomics closer to practical deployment.

## 7 ACKNOWLEDGEMENT

This work is partly supported by the Federal Ministry of Research, Technology and Space in DAAD project 57616814 (SECAI, School of Embedded Composite AI, https://secai.org/) as part of the program Konrad Zuse Schools of Excellence in Artificial Intelligence.

## 8 CONFLICT OF INTEREST

JNK declares ongoing consulting services for AstraZeneca and Bioptimus. Furthermore, he holds shares in StratifAI, Synagen, and Spira Labs, has received an institutional research grant from GSK and AstraZeneca, as well as honoraria from AstraZeneca, Bayer, Daiichi Sankyo, Eisai, Janssen, Merck, MSD, BMS, Roche, Pfizer, and Fresenius.

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

## A    APPENDIX: THE USE OF LARGE LANGUAGE MODELS

This work benefited from the use of Large Language Models (LLMs) for minor tasks such as text and language polishing, as well as for suggesting memorable model names. LLMs were not used to generate scientific content, derive conclusions, or perform any form of data analysis. The authors are fully responsible for the entire content and integrity of this submission.

## B    APPENDIX: CODE, TRAINING CONFIGURATION, AND DATA SPLITS

All code, training configurations, and data splits are provided for reproducibility. The HEST1k dataset was obtained following the original publication (Jaume et al., 2024), and PFM preprocessing followed official repositories (Chen et al., 2024b; Xu et al., 2024). Baseline models were implemented according to their official repositories(Huang et al., 2025b; Zhu et al., 2025). Scripts will be released upon acceptance.

### B.1    IMPLEMENTATION AND EVALUATION DETAILS

Models were trained end-to-end with MSE loss using the AdamW optimizer with learning rate $5 \times 10^{-4}$ and weight decay 0.01. Training was run for up to 1000 epochs with early stopping patience of 30 epochs based on validation MSE, while convergence is usually achieved after approximately 300 epochs. Gradient clipping with a maximum norm of 1.0 was applied. Sample sizes for two splits are shown in Table 5. Sample size for each cancer type in test splits are shown in Table 6. All experiments used PyTorch on a single NVIDIA A100 GPU.

Table 5: Number of samples in splits.

| Splits | #Train | #Validation | #Test |
|--------|--------|-------------|-------|
| Split 0 | 501 | 124 | 23 |
| Split 1 | 498 | 123 | 28 |

Table 6: Test set sample sizes across 10 cancer types in two splits.

| Cancer Type | # Samples | | |
|-------------|-----------|---------|-------|
| | Split 0 | Split 1 | Total |
| CCRCC | 4 | 4 | 8 |
| COAD | 3 | 1 | 4 |
| HCC | 1 | 1 | 2 |
| IDC | 1 | 1 | 2 |
| LUNG | 1 | 1 | 2 |
| LYMPH_IDC | 1 | 1 | 2 |
| PAAD | 1 | 1 | 2 |
| PRAD | 8 | 15 | 23 |
| READ | 2 | 2 | 4 |
| SKCM | 1 | 1 | 2 |
| Total | 23 | 28 | 51 |

## C    APPENDIX: GENE PATHWAY COHERENCE DETAILS

We compute the variance of each gene across the test set for two splits and discretize them into ten variance levels (1–10). For each pathway, we then calculate the unweighted average variance of its constituent genes to derive the pathway-level variance. The variance levels are summarized in Table 7, while Figure 5 provides a more intuitive visualization of the distribution. Gene counts and average variances per pathway are reported in Table 8  9.

Table 7: Gene variance level thresholds details.

| Levels | Variance Threshold | |
| --- | --- | --- |
| | Split 0 | Split 1 |
| Level 1 | > 0.0000 | >0.0000 |
| Level 2 | > 0.7525 | >0.7614 |
| Level 3 | > 0.8663 | >0.8462 |
| Level 4 | > 0.9070 | >0.8865 |
| Level 5 | > 0.9360 | >0.9213 |
| Level 6 | > 0.9662 | >0.9609 |
| Level 7 | > 1.0079 | >1.0145 |
| Level 8 | > 1.0698 | >1.0899 |
| Level 9 | > 1.1764 | >1.2015 |
| Level 10 | > 1.3743 | >1.3806 |

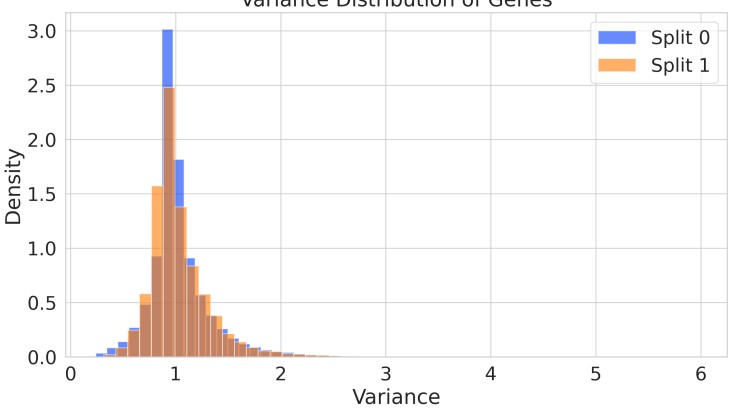

Figure 5: Gene variance distribution density plot.

Table 8: Comparison of Hallmark pathway-level PCC across models.

| Pathway ID | #Genes | Avg. Variance | Variance Level | PCC | |
|---|---|---|---|---|---|
| | | | | **HistoPrism** | **STPath** |
| HALLMARK_ADIPOGENESIS | 60 | 1.0210 | 6.5 | 0.2597 | 0.2590 |
| HALLMARK_ALLOGRAFT_REJECTION | 67 | 1.3370 | 9.5 | 0.2292 | 0.2291 |
| HALLMARK_ANDROGEN_RESPONSE | 36 | 1.0500 | 7.5 | 0.3199 | 0.2907 |
| HALLMARK_ANGIOGENESIS | 16 | 1.2810 | 9.0 | 0.2267 | 0.2239 |
| HALLMARK_APICAL_JUNCTION | 71 | 1.1703 | 8.5 | 0.2488 | 0.2534 |
| HALLMARK_APICAL_SURFACE | 16 | 1.2029 | 8.5 | 0.2222 | 0.2292 |
| HALLMARK_APOPTOSIS | 79 | 1.1056 | 7.5 | 0.2716 | 0.2613 |
| HALLMARK_BILE_ACID_METABOLISM | 28 | 1.0077 | 6.0 | 0.2136 | 0.1713 |
| HALLMARK_CHOLESTEROL_HOMEOSTASIS | 31 | 1.0467 | 7.0 | 0.2660 | 0.2774 |
| HALLMARK_COAGULATION | 33 | 1.2280 | 8.5 | 0.2279 | 0.2257 |
| HALLMARK_COMPLEMENT | 62 | 1.2166 | 8.5 | 0.2423 | 0.2406 |
| HALLMARK_DNA_REPAIR | 40 | 0.9847 | 5.5 | 0.2738 | 0.2627 |
| HALLMARK_E2F_TARGETS | 56 | 1.0199 | 6.5 | 0.2973 | 0.2820 |
| HALLMARK_EPITHELIAL_MESENCHYMAL_TRANSITION | 83 | 1.3331 | 9.5 | 0.2288 | 0.2286 |
| HALLMARK_ESTROGEN_RESPONSE_EARLY | 53 | 1.1473 | 8.0 | 0.2729 | 0.2612 |
| HALLMARK_ESTROGEN_RESPONSE_LATE | 55 | 1.2174 | 8.5 | 0.2978 | 0.2781 |
| HALLMARK_FATTY_ACID_METABOLISM | 35 | 1.0369 | 7.0 | 0.2748 | 0.2614 |
| HALLMARK_G2M_CHECKPOINT | 64 | 1.0719 | 7.5 | 0.3008 | 0.2816 |
| HALLMARK_GLYCOLYSIS | 47 | 1.0945 | 7.5 | 0.2995 | 0.2735 |
| HALLMARK_HEDGEHOG_SIGNALING | 11 | 1.1500 | 8.0 | 0.2198 | 0.1955 |
| HALLMARK_HEME_METABOLISM | 48 | 1.0743 | 7.5 | 0.2678 | 0.2446 |
| HALLMARK_HYPOXIA | 71 | 1.1004 | 8.0 | 0.2717 | 0.2586 |
| HALLMARK_IL2_STAT5_SIGNALING | 75 | 1.1960 | 8.5 | 0.2505 | 0.2330 |
| HALLMARK_IL6_JAK_STAT3_SIGNALING | 43 | 1.3526 | 9.5 | 0.2121 | 0.2101 |
| HALLMARK_INFLAMMATORY_RESPONSE | 76 | 1.3428 | 9.5 | 0.2070 | 0.2115 |
| HALLMARK_INTERFERON_ALPHA_RESPONSE | 34 | 1.2280 | 9.0 | 0.2306 | 0.2185 |
| HALLMARK_INTERFERON_GAMMA_RESPONSE | 77 | 1.2545 | 9.0 | 0.2166 | 0.2205 |
| HALLMARK_KRAS_SIGNALING_DN | 20 | 1.2615 | 9.0 | 0.1836 | 0.1787 |
| HALLMARK_KRAS_SIGNALING_UP | 70 | 1.3569 | 9.5 | 0.2309 | 0.2314 |
| HALLMARK_MITOTIC_SPINDLE | 56 | 1.0445 | 6.5 | 0.2795 | 0.2542 |
| HALLMARK_MTORC1_SIGNALING | 67 | 0.9656 | 5.0 | 0.3044 | 0.2963 |
| HALLMARK_MYC_TARGETS_V1 | 62 | 0.9274 | 4.5 | 0.3343 | 0.2981 |
| HALLMARK_MYC_TARGETS_V2 | 15 | 0.8886 | 3.0 | 0.1711 | 0.1145 |
| HALLMARK_MYOGENESIS | 52 | 1.1201 | 8.0 | 0.2510 | 0.2558 |
| HALLMARK_NOTCH_SIGNALING | 16 | 1.1593 | 8.0 | 0.2535 | 0.2452 |
| HALLMARK_OXIDATIVE_PHOSPHORYLATION | 41 | 0.8582 | 2.5 | 0.2435 | 0.1985 |
| HALLMARK_P53_PATHWAY | 66 | 1.1311 | 8.0 | 0.2665 | 0.2658 |
| HALLMARK_PANCREAS_BETA_CELLS | 8 | 1.4304 | 10.0 | 0.1312 | 0.1069 |
| HALLMARK_PEROXISOME | 32 | 1.0145 | 7.0 | 0.2903 | 0.2691 |
| HALLMARK_PI3K_AKT_MTOR_SIGNALING | 51 | 1.0554 | 7.0 | 0.2813 | 0.2733 |
| HALLMARK_PROTEIN_SECRETION | 33 | 0.9266 | 4.5 | 0.3179 | 0.2979 |
| HALLMARK_REACTIVE_OXYGEN_SPECIES_PATHWAY | 15 | 0.9336 | 4.5 | 0.1843 | 0.1383 |
| HALLMARK_SPERMATOGENESIS | 16 | 1.0683 | 7.5 | 0.2630 | 0.2342 |
| HALLMARK_TGF_BETA_SIGNALING | 30 | 1.1092 | 7.5 | 0.2610 | 0.2466 |
| HALLMARK_TNFA_SIGNALING_VIA_NFKB | 71 | 1.2954 | 9.0 | 0.2369 | 0.2269 |
| HALLMARK_UNFOLDED_PROTEIN_RESPONSE | 24 | 0.9069 | 4.0 | 0.2835 | 0.2455 |
| HALLMARK_UV_RESPONSE_DN | 65 | 1.1875 | 8.5 | 0.2340 | 0.2265 |
| HALLMARK_UV_RESPONSE_UP | 50 | 1.1027 | 7.5 | 0.3034 | 0.2699 |
| HALLMARK_WNT_BETA_CATENIN_SIGNALING | 21 | 1.1335 | 8.0 | 0.2284 | 0.2193 |
| HALLMARK_XENOBIOTIC_METABOLISM | 53 | 1.0462 | 7.0 | 0.2771 | 0.2505 |

Table 9: Comparison of GO pathway-level PCC across models.

| Pathway ID | #Genes | Avg. Variance | Variance Level | PCC HistoPrism | STPath |
|---|---|---|---|---|---|
| GOBP_ACTIVATION_OF_INNATE_IMMUNE_RESPONSE | 100 | 1.0227 | 6.5 | 0.2364 | 0.2139 |
| GOBP_B_CELL_ACTIVATION | 100 | 1.1720 | 8.5 | 0.2325 | 0.2218 |
| GOBP_CALCIUM_ION_TRANSPORT | 100 | 1.1175 | 7.5 | 0.2446 | 0.2314 |
| GOBP_NEURON_APOPTOTIC_PROCESS | 100 | 1.0951 | 7.5 | 0.2545 | 0.2654 |
| GOBP_REGULATION_OF_CELLULAR_RESPONSE_TO_GROWTH_FACTOR_STIMULUS | 100 | 1.1876 | 8.5 | 0.2165 | 0.2270 |
| GOCC_COLLAGEN_CONTAINING_EXTRACELLULAR_MATRIX | 100 | 1.3604 | 9.5 | 0.1953 | 0.2048 |
| GOCC_GOLGI_MEMBRANE | 100 | 1.0275 | 7.0 | 0.2600 | 0.2531 |
| GOCC_MEMBRANE_MICRODOMAIN | 100 | 1.1672 | 8.5 | 0.2520 | 0.2464 |
| GOBP_CELL_PROJECTION_ASSEMBLY | 99 | 1.0780 | 7.5 | 0.2317 | 0.2237 |
| GOBP_LIPID_LOCALIZATION | 99 | 1.0686 | 7.5 | 0.2170 | 0.2125 |
| GOBP_MACROAUTOPHAGY | 99 | 0.9678 | 6.0 | 0.2099 | 0.2130 |
| GOCC_PRESYNAPSE | 99 | 0.9804 | 6.0 | 0.2710 | 0.2468 |
| GOBP_REGULATION_OF_ENDOCYTOSIS | 98 | 1.1148 | 8.0 | 0.2341 | 0.2303 |
| GOBP_RIBONUCLEOPROTEIN_COMPLEX_BIOGENESIS | 97 | 0.9136 | 4.0 | 0.2035 | 0.1648 |
| GOBP_ORGANOPHOSPHATE_BIOSYNTHETIC_PROCESS | 96 | 1.0071 | 6.5 | 0.2882 | 0.2728 |
| GOBP_REGULATION_OF_NEUROGENESIS | 96 | 1.1289 | 8.0 | 0.2237 | 0.2246 |
| GOMF_SIGNALING_RECEPTOR_REGULATOR_ACTIVITY | 96 | 1.3151 | 9.0 | 0.1967 | 0.1966 |
| GOBP_RHYTHMIC_PROCESS | 95 | 1.0576 | 7.0 | 0.2516 | 0.2513 |
| GOCC_NUCLEAR_SPECK | 95 | 0.9567 | 5.5 | 0.2261 | 0.2274 |
| GOMF_AMINOACYLTRANSFERASE_ACTIVITY | 95 | 0.9781 | 6.0 | 0.2969 | 0.2799 |
| GOBP_NEGATIVE_REGULATION_OF_MOLECULAR_FUNCTION | 94 | 1.1241 | 8.0 | 0.2465 | 0.2355 |
| GOBP_RESPONSE_TO_ALCOHOL | 94 | 1.1364 | 8.0 | 0.2609 | 0.2516 |
| GOMF_GUANYL_NUCLEOTIDE_BINDING | 94 | 1.0072 | 6.0 | 0.2679 | 0.2502 |
| GOBP_EPIDERMIS_DEVELOPMENT | 93 | 1.3638 | 9.5 | 0.2616 | 0.2586 |
| GOBP_POSITIVE_REGULATION_OF_PHOSPHORYLATION | 93 | 1.1214 | 8.0 | 0.2332 | 0.2260 |
| GOBP_NEGATIVE_REGULATION_OF_CELL_CYCLE | 92 | 1.1207 | 8.0 | 0.2387 | 0.2418 |
| GOBP_MALE_GAMETE_GENERATION | 90 | 1.0272 | 7.0 | 0.2438 | 0.2316 |
| GOBP_NEGATIVE_REGULATION_OF_CYTOKINE_PRODUCTION | 90 | 1.1553 | 8.0 | 0.2180 | 0.2130 |
| GOBP_CELLULAR_RESPONSE_TO_INSULIN_STIMULUS | 89 | 1.0354 | 7.0 | 0.2296 | 0.2204 |
| GOBP_MEMBRANELESS_ORGANELLE_ASSEMBLY | 89 | 0.9689 | 5.5 | 0.2445 | 0.2484 |
| GOBP_REGULATION_OF_TRANSLATION | 89 | 0.9687 | 5.5 | 0.2842 | 0.2682 |
| GOMF_ACTIN_BINDING | 88 | 1.0558 | 7.0 | 0.2709 | 0.2615 |
| GOMF_PHOSPHOLIPID_BINDING | 88 | 0.9990 | 6.0 | 0.2476 | 0.2410 |
| GOMF_TRANSCRIPTION_COACTIVATOR_ACTIVITY | 88 | 0.9709 | 6.0 | 0.2685 | 0.2424 |
| GOCC_MICROTUBULE | 86 | 1.0171 | 6.5 | 0.2903 | 0.2666 |
| GOCC_MITOCHONDRIAL_MATRIX | 86 | 0.9383 | 4.5 | 0.2907 | 0.2763 |
| GOMF_PHOSPHORIC_ESTER_HYDROLASE_ACTIVITY | 85 | 1.0700 | 7.5 | 0.2471 | 0.2427 |
| GOCC_TRANSFERASE_COMPLEX_TRANSFERRING_PHOSPHORUS_CONTAINING_GROUPS | 83 | 0.9975 | 6.0 | 0.2846 | 0.2660 |
| GOBP_PROTEIN_LOCALIZATION_TO_CELL_PERIPHERY | 82 | 0.9707 | 5.5 | 0.2842 | 0.2675 |
| GOCC_BASAL_PART_OF_CELL | 82 | 1.1528 | 8.5 | 0.2798 | 0.2724 |
| GOMF_ENZYME_INHIBITOR_ACTIVITY | 82 | 1.1447 | 8.0 | 0.2280 | 0.2239 |
| GOBP_PROTEIN_LOCALIZATION_TO_EXTRACELLULAR_REGION | 80 | 1.0677 | 7.0 | 0.2251 | 0.2210 |
| GOBP_POSITIVE_REGULATION_OF_CELL_CYCLE | 79 | 1.1142 | 7.5 | 0.2598 | 0.2419 |
| GOBP_AMEBOIDAL_TYPE_CELL_MIGRATION | 78 | 1.1797 | 8.5 | 0.2417 | 0.2416 |
| GOBP_CELLULAR_RESPONSE_TO_RADIATION | 78 | 1.0697 | 7.5 | 0.2265 | 0.2304 |
| GOBP_REGULATION_OF_INTRACELLULAR_TRANSPORT | 78 | 1.0626 | 7.5 | 0.2655 | 0.2497 |
| GOCC_NUCLEAR_MEMBRANE | 78 | 1.0141 | 6.5 | 0.2660 | 0.2545 |
| GOCC_VESICLE_LUMEN | 78 | 1.0847 | 7.5 | 0.2537 | 0.2472 |
| GOBP_REGULATION_OF_MYELOID_CELL_DIFFERENTIATION | 77 | 1.2347 | 8.5 | 0.2274 | 0.2239 |
| GOBP_FAT_CELL_DIFFERENTIATION | 75 | 1.1462 | 8.0 | 0.1980 | 0.2062 |
| GOBP_REGULATION_OF_PROTEOLYSIS_INVOLVED_IN_PROTEIN_CATABOLIC_PROCESS | 75 | 0.9792 | 6.0 | 0.2533 | 0.2440 |
| GOCC_SECRETORY_GRANULE_MEMBRANE | 75 | 1.2631 | 8.5 | 0.2648 | 0.2551 |
| GOBP_TISSUE_HOMEOSTASIS | 74 | 1.2416 | 9.0 | 0.2463 | 0.2481 |
| GOBP_RESPONSE_TO_MECHANICAL_STIMULUS | 73 | 1.2086 | 8.5 | 0.2233 | 0.2183 |
| GOMF_CATALYTIC_ACTIVITY_ACTING_ON_DNA | 73 | 0.9673 | 5.5 | 0.2535 | 0.2400 |
| GOMF_PHOSPHATASE_BINDING | 71 | 1.1472 | 8.0 | 0.2587 | 0.2621 |
| GOBP_ALCOHOL_METABOLIC_PROCESS | 70 | 1.0000 | 6.0 | 0.2388 | 0.2239 |
| GOMF_PROTEIN_HETERODIMERIZATION_ACTIVITY | 70 | 1.0976 | 7.5 | 0.2687 | 0.2690 |
| GOBP_COGNITION | 69 | 1.0499 | 7.0 | 0.2259 | 0.2274 |
| GOMF_SULFUR_COMPOUND_BINDING | 69 | 1.3182 | 9.5 | 0.2036 | 0.2084 |
| GOBP_JNK_CASCADE | 68 | 1.0452 | 7.0 | 0.2354 | 0.2201 |
| GOBP_PROTEIN_COMPLEX_OLIGOMERIZATION | 68 | 1.0248 | 7.0 | 0.2520 | 0.2359 |
| GOBP_CARBOHYDRATE_DERIVATIVE_CATABOLIC_PROCESS | 67 | 1.0404 | 7.0 | 0.2002 | 0.1878 |
| GOBP_PROTEIN_PROCESSING | 67 | 1.0879 | 7.5 | 0.2410 | 0.2302 |
| GOMF_DNA_BINDING_TRANSCRIPTION_REPRESSOR_ACTIVITY | 65 | 1.1279 | 8.0 | 0.2288 | 0.2167 |
| GOBP_ORGANIC_ACID_BIOSYNTHETIC_PROCESS | 64 | 1.0790 | 7.5 | 0.2436 | 0.2270 |
| GOBP_REGULATION_OF_EXTRINSIC_APOPTOTIC_SIGNALING_PATHWAY | 64 | 1.1024 | 7.5 | 0.2235 | 0.2298 |
| GOCC_POSTSYNAPTIC_SPECIALIZATION | 63 | 1.0164 | 7.0 | 0.2580 | 0.2416 |
| GOBP_CARTILAGE_DEVELOPMENT | 62 | 1.2839 | 9.0 | 0.2165 | 0.2282 |
| GOBP_ADAPTIVE_THERMOGENESIS | 61 | 1.1568 | 8.0 | 0.2349 | 0.2429 |
| GOBP_GLAND_MORPHOGENESIS | 61 | 1.2464 | 9.0 | 0.2516 | 0.2470 |
| GOBP_REGULATION_OF_REACTIVE_OXYGEN_SPECIES_METABOLIC_PROCESS | 61 | 1.0773 | 7.5 | 0.2501 | 0.2394 |
| GOCC_PLASMA_MEMBRANE_SIGNALING_RECEPTOR_COMPLEX | 60 | 1.1848 | 8.5 | 0.2476 | 0.2481 |
| GOBP_ENDOTHELIUM_DEVELOPMENT | 59 | 1.2269 | 9.0 | 0.2375 | 0.2355 |
| GOCC_COATED_VESICLE | 58 | 1.0659 | 7.5 | 0.2823 | 0.2632 |
| GOBP_REGULATION_OF_CELL_MATRIX_ADHESION | 57 | 1.2320 | 8.5 | 0.2493 | 0.2466 |
| GOCC_CYTOPLASMIC_SIDE_OF_MEMBRANE | 57 | 1.1110 | 8.0 | 0.2631 | 0.2553 |
| GOBP_ANATOMICAL_STRUCTURE_MATURATION | 56 | 1.1621 | 8.0 | 0.2311 | 0.2359 |
| GOBP_CELLULAR_RESPONSE_TO_METAL_ION | 56 | 1.0699 | 7.5 | 0.2679 | 0.2571 |
| GOBP_NEGATIVE_REGULATION_OF_GENE_EXPRESSION_EPIGENETIC | 56 | 0.9559 | 5.0 | 0.1923 | 0.1510 |
| GOBP_ESTABLISHMENT_OF_CELL_POLARITY | 54 | 1.0476 | 7.0 | 0.2370 | 0.2313 |
| GOBP_REGULATION_OF_CELL_MORPHOGENESIS | 54 | 1.0163 | 6.5 | 0.2193 | 0.2087 |
| GOBP_RESPONSE_TO_TOPOLOGICALLY_INCORRECT_PROTEIN | 54 | 0.9582 | 5.0 | 0.2880 | 0.2740 |
| GOBP_SENSORY_PERCEPTION | 54 | 1.1405 | 8.0 | 0.2216 | 0.2162 |
| GOMF_CARBOHYDRATE_BINDING | 54 | 1.1295 | 8.0 | 0.2071 | 0.2134 |
| GOBP_REGULATION_OF_RESPONSE_TO_WOUNDING | 52 | 1.2147 | 8.5 | 0.2394 | 0.2385 |
| GOBP_CELL_CELL_ADHESION_VIA_PLASMA_MEMBRANE_ADHESION_MOLECULES | 51 | 1.2749 | 9.0 | 0.2401 | 0.2455 |

