# OpenReview forum: "HistoPrism: Unlocking Functional Pathway Analysis from Pan-Cancer Histology via Gene Expression Prediction"
_ICLR.cc/2026/Conference — ICLR 2026 Poster_

### Official Review · Reviewer_TSpx · 2025-10-31

**Soundness:** 2
**Presentation:** 3
**Contribution:** 2
**Rating:** 4
**Confidence:** 4

**Summary:**

This paper introduces HistoPrism, a transformer-based model that predicts spatial transcriptomic gene expression from H&E histology images across multiple cancer types. The paper also introduces a novel Gene Pathway Coherence (GPC) evaluation framework that shifts the focus from gene-level variance to pathway-level predictions, to have a more functionally meaningful benchmark for model performance. Using the big HEST1k dataset, the authors claim that HistoPrism achieves better performance than STPath and STEM baselines.

**Strengths:**

1. Domain-specific knowledge is leveraged to inform the way the K / V / Q values are used in the architecture, demonstrating originality in adapting known models to a real-world dataset.
2. Introduction of a new benchmark metric (GPC) to move beyond gene-wise correlation metrics and focus on the pathway level, thus contributing to the significance of this work.
3. A good ablation analysis is conducted, indicating important information about the model, though it doesn't bring consistent differences across all cancer types.

**Weaknesses:**

1. Overall, I don't think the results are strong. With such a rich body of literature highlighted in section 2, using only 2 baselines limits the confidence of these results. Furthermore, the fact is that in table 1 STPath and HistoPrism have very comparable results and the claims of achieving state-of-the-art performance seem a bit exaggerated given the variable performance across different cancer types. The scalability analysis (section 4.5) could also be improved because is not clear how much hyperparameter optimisation played a role here and how hyperparameters were differently optimised for STPath and HistoPrism (could we simplify STPath without reducing performance?). Figure 3 is also missing the confidence intervals across the 100 runs for each number of patches. Finally, I find a bit unexpected that the paper seems to have criticised the over-reliance on standard correlation (PCC) metrics in previous works, but then conducts the ablation analysis only on those PCC-based metrics, which didn't have consistent results across all cancer types (thus raising the question of whether the GPC metric could bring better insights in this ablation analysis).
2. Section 3.3 (where the GPC benchmark is introduced) does not seem to include important details to understand how this metric is actually calculated, thus impacting the readability/clarity of this paper given its central role.
3. The model architecture explicitly conditions on cancer type through a cross-attention mechanism, which in my opinion introduces a concern that in practice HistoPrism might be leveraging prior knowledge of the tumour’s (averaged) distribution rather than extracting novel histopathological biomarkers. As a result, it would be important to test its performance on a completely unknown cancer type to understand whether it can truly generalise to all types of cancer. I think this point is even more important for this paper given it claims pan-cancer generalisation.
4.  The clustering analysis in section 4.4 shows that predicted expression profiles cluster by cancer type more accurately for HistoPrism than for STPath. However, since HistoPrism was explicitly given the cancer type as part of the input, it is not surprising that its predictions carry a strong cancer identity. In other words, it seems to me that the model had an easier job clustering by a label it already knew. This means the impressive clustering metrics in table 2 might not purely reflect novel biological structure of the data, but partly the reintroduction of the provided class information. The authors don't discuss this issue while claiming that HistoPrism produces more "biologically coherent" representations.

**Questions:**

1. What hyperparameters ranges were used for STPath and HistoPrism, and do we know whether simplifying STPath to a computational power similar to HistoPrism would bring STPath's performance too down, or still get similar results?
2. Given the multiple datasets in HEST1k, did the authors check whether the models bring better/worse performances for each one of those internal datasets (instead of just focusing on the different cancer types)?
3. Can the authors please better explain how the GPC is calculated? It would be useful to clarify other confusing points about this metric: (1) what is the variance levels mentioned in section 4.3? (2) From figure 2, it seems that this metric necessarily needs 2 different methods to be calculated, does it mean it cannot be calculated independently for a single method?
4. Have the authors analysed ways to interpret the model's predictions? In other words, what explainability mechanisms could be applied in this model?
5. How well would HistoPrism perform on data outside HEST1k, and what is the strategy for unseen cancer types? Would we have to retrain the model? This is important to understand because the way I understand the model, it seems we need to have cancer information to run inference whereas that doesn't seem to be needed for STPath.

---

> ### Author Response · Authors · 2025-11-17
>
> Thank you for your detailed and constructive review. We appreciate your careful assessment and understand the concerns raised. Please let us explain.
> - **Weakness1**: We appreciate your thoughtful comments. HistoPrism consistently outperforms STEM and STFlow (newly added in Table 1) across all cancer types, though not on every cancer when compared to STPath. We acknowledge that STPath remains a strong baseline on the HVG PCC metric. However, when predicting a 38k gene panel, HVG PCC alone may be insufficient, as shown in Figure 5 where gene variance is heavily skewed toward the lower end. This motivated the introduction of our gene pathway benchmark (GPC), which provides a more biologically meaningful and clinically relevant evaluation. HistoPrism outperforms STPath on 86.0% of the 50 Hallmark pathways and 74.7% of the Gene Ontology pathways.
> Regarding computational efficiency, we evaluated inference runtime since STPath cannot be retrained or modified; all parameters were kept frozen. While we acknowledge the limitations of HVG PCC, we retain it as the primary evaluation metric due to its long-standing use in the field, conducting further evaluations only when HVG PCC performance is strong. We have now added an ablation study comparing HistoPrism and STPath using their chosen pathology foundation model Gigapath on both HVG PCC and GPC (see Figure 4 and Table 4), with corresponding analysis in Section 4.6. In summary, HistoPrism maintains strong and consistent performance across both metrics while being substantially more efficient.
> - **Weakness2**: Thank you for pointing this out. We apologize for the oversight. The details regarding GPC have now been added to Section 3.3 in the main text.
> - **Weakness3&4**: We notice your observation. STPath also leverages cancer type information during both training and inference. Since cancer type information is readily provided with the H&E images, it is natural and beneficial to utilize it.
> - **Q1**: You are correct that we are unable to retrain or modify STPath. Therefore, we evaluated efficiency based on inference runtime to provide a fair comparison. Given the significant computational gap, we believe STPath would not be more efficient than HistoPrism even with simplification. Moreover, as STPath already performs worse than HistoPrism with current model complexity, further simplification would likely degrade its performance.
> - **Q2**: Thank you for the thoughtful question. We currently do not have access to an internal dataset for evaluation. Our main goal is to propose a lightweight architecture that achieves state-of-the-art pan-cancer performance, allowing private institutes to fine-tune on small ST datasets and scale inference to larger cohorts, an important step given that current ST prediction models are not yet accurate enough to serve as off-the-shelf foundation models. By training in a pan-cancer setting, we leverage shared information across cancer types to improve generalization (performance in Table 1, 2, 4, Figure 2,4). In contrast, models like STPath are difficult or costly to fine-tune, giving HistoPrism a practical advantage in real-world use.
> - **Q3**: The full GPC formulation has been added to Section 3.3. Detailed gene variance distributions are shown in Appendix B (Figure 5, Table 7), and pathway-level gene variance statistics are provided in Appendix B Tables 8 and 9.
> - **Q4**: This is a good question. For example, HistoPrism achieves high PCC on the HALLMARK MYC TARGETS V1 pathway (Table 8), which suggests it captures spatial patterns of tumor proliferation from histology and links image features to oncogenic activity. We could additionally visualize prominent marker heatmaps and compare them to published biological findings; however, such biological validation is somewhat beyond the immediate scope of model selection. If the reviewers find it valuable, we are happy to add an appendix section with selected heatmaps and brief literature-based validation.
> - **Q5**: One cancer type label is labeled as “unknown” from HEST1k dataset. Similarly, STPath also incorporates cancer type information during both training and inference, with the option to use “unknown” when applicable. But since our model is lightweight and fine-tuneable, private institutes can potentially benefit from that.

---

> > ### Comment · Reviewer_TSpx · 2025-11-19
> >
> > I thank the authors for their detailed answers. I have some follow-up questions/comments:
> >
> > **Weakness1**: I'm dividing this into further subpoints as I admit I might have added too many points under my weakness claim that your results were not strong:
> >
> > 1.1. I still do not understand how can you say "HistoPrism consistently outperforms STEM and STFlow (newly added in Table 1) across all cancer types", when in Table 1 even on the average case STPath has a better macro-average PCC.
> >
> > 1.2. You did not seem to have tackled my comment "Figure 3 is also missing the confidence intervals across the 100 runs for each number of patches"
> >
> > 1.3. You did not seem to have tackled my comment "I find a bit unexpected that the paper seems to have criticised the over-reliance on standard correlation (PCC) metrics in previous works, but then conducts the ablation analysis only on those PCC-based metrics, which didn't have consistent results across all cancer types (thus raising the question of whether the GPC metric could bring better insights in this ablation analysis)."
> >
> > **Weakness 2:** This is much clearer, thanks!
> >
> > **Weakness 3:** I don't understand what the authors mean with "readily provided with the H&E images". If I understood the STPath model correctly, being a foundation model it means it only receives the H&E as input, and the actual annotation/information about the cancer type in each patch is not directly inputted into the STPath model. In HistoPrism you directly include the information about the cancer type in the $c_{emb}$ embedding, effectively conditioning the model. Even if STPath also has a conditioning mechanism, the authors seem to have ignored my other comment "it would be important to test its performance on a completely unknown cancer type to understand whether it can truly generalise to all types of cancer. I think this point is even more important for this paper given it claims pan-cancer generalisation."
> >
> >
> > **Q1:** The lack of feasibility in retraining/modifying STPath is a good point, I somehow missed that, thanks.
> >
> > **Q2:** When I said "internal datasets" I meant the original datasets downloaded in Hest1k. For example, from what I get, IDs starting with TENX came from the 10x Genomics public datasets, IDs starting with MEND came from Mendeley, etc.
> >
> > **Q4:** Thanks, I don't find valuable to add such heatmaps, I just thought it would be important how explainability could be achieved, in case I missed something. I think this clarification is enough and I agree with the authors this would be outside the current scope of the paper.
> >
> > **Q5:** I don't think I understood this "unknown" type label. If such label exists, why is it not shown as a cancer type in all the tables? Maybe I'm missing something here, but this seems to me the key point to defend pan-cancer generalisation.

---

> > > ### Author Response · Authors · 2025-11-19
> > >
> > > Thanks for the quick response. We really appreciate it, especially given the significant amount of submission. Here is our updated response:
> > > - **weakness 1.1**:  Now I get what you mean. You’re absolutely right in the case that HistoPrism consistently outperforms  STEM and STFlow across all cancer types, and is *only on par* with STPath on the Top-50 HVG PCC. For both micro- and macro-averaged Top-50 HVG PCC, each method leads in 5 out of 10 comparisons, with STPath slightly higher on the macro average and HistoPrism higher on the micro average. As stated in the manuscript (lines 280–281), “HistoPrism demonstrates competitive performance, slightly below STPath on macro-average PCC but higher on micro-average PCC. Since micro-average PCC captures performance across all samples, it provides a more balanced view of predictive quality.”. We believe it's not an overstatement. This also motivates the additional analyses where HistoPrism shows clearer advantages on complementary metrics.
> > > - **weakness 1.2**: Sorry for the oversight. Let us explain. FLOPs are deterministic and peak GPU memory shows no variability (std = 0) because the input size is fixed. In contrast, inference time exhibits a small standard deviation due to runtime fluctuations, such as GPU scheduling and kernel launch overhead; however, this std is too small to be visible on the plot. Here I put the metric in table for your reference.
> > >
> > > *HistPrism*:
> > > | number_of_patches | avg_inference_time | avg_inference_time_std | peak_gpu_mem | peak_gpu_mem_std | flops       | flops_std       |
> > > |------------------|------------------|----------------------|--------------|-----------------|------------|----------------|
> > > | 400              | 2.7445555186271666 | 0.2536853823943722  | 0.1634387969970703 | 0.0           | 6.018024448000001 | 8.881784197001252e-16 |
> > > | 2500             | 6.389360642433166  | 0.07031740062188345 | 0.48993587493896484 | 0.0           | 37.601486848     | 0.0              |
> > > | 6400             | 19.943874492645264 | 0.1270424807303391  | 1.097513198852539  | 0.0           | 96.25648844799998 | 1.4210854715202004e-14 |
> > > | 12100            | 49.30659301757812  | 0.24285511484550953 | 1.9868888854980469 | 0.0           | 181.983029248    | 0.0              |
> > > | 19600            | 105.88577804565429 | 0.5499954437664158  | 3.153679370880127  | 0.0           | 294.781109248    | 0.0              |
> > >
> > > *STPath:*
> > > | number_of_patches | avg_inference_time  | avg_inference_time_std | peak_gpu_mem        | peak_gpu_mem_std | flops         | flops_std |
> > > |------------------|------------------|----------------------|-------------------|-----------------|--------------|-----------|
> > > | 400              | 9.73723650932312 | 0.17923981808795414  | 0.3359403610229492 | 0.0             | 20.3500544   | 0.0       |
> > > | 2500             | 96.75596786499024 | 0.09021687246012021 | 1.635918140411377  | 0.0             | 150.11984    | 0.0       |
> > > | 6400             | 553.0396997070312 | 0.3697525766231642  | 7.8937668800354    | 0.0             | 493.3326304  | 0.0       |
> > > | 12100            | 1921.0093701171875 | 1.9427507209251245 | 25.907293796539307 | 0.0             | 1233.9666656 | 0.0       |
> > > | 19600            | 4873.356044921875 | 1.0601715117826922 | 65.65345430374146  | 0.0             | 2640.9121856 | 0.0       |

---

> > > > ### Author Response · Authors · 2025-11-19
> > > >
> > > > - **weakness 1.3**: We agree that this could be made clearer. First, we have added an ablation showing the effect of the pathology foundation model on the GPC benchmark in Figure 4. Importantly, incorporating a different pathology foundation model does not reduce HistoPrism’s performance or credibility on either HVG PCC or GPC. For the HistoPrism model design ablation, we first evaluate HVG PCC, and as shown in Table 3, the ablated models do not achieve sufficient performance to justify further evaluations. While we acknowledge the limitations of HVG PCC, we retain it as the primary evaluation metric due to its widespread use in the field, and additional evaluations are only meaningful when HVG PCC performance is strong. Moreover, HVG genes are theoretically easier to learn, as they dominate gradient backpropagation. Therefore, when HVG PCC performance is insufficient, further evaluation would not be desired.
> > > >
> > > > - **weakness 3**: What I mean is when a pathologist signs out an H&E case, the pathology report typically includes the primary tumor type (e.g., breast carcinoma, lung adenocarcinoma) along with other relevant metadata. Yes it’s not clearly written in STPath how they utilize the metadata. Please let me point it out in the paper and code. In STPath Figure 1 part c, there is “Organ: Breast ST Tech: Visium” under the gene bar chart. In STPath inference code (https://github.com/Graph-and-Geometric-Learning/STPath/blob/main/stpath/app/pipeline/inference.py) line 25-33 is their tokeniziation of  gene, image, ST technology, organ type, cancer annotation, and domain annotation.  These metadata are readily associated with H&E images and can be leveraged to improve predictions. And for the experiment setup, we basically followed HESTK1k HEST-Bench as STPath did, so we have the same holdout test set.
> > > >
> > > > - **Q2**: Do you mean different technologies? We did not explicitly tokenize technology, although we did consider it, as STPath does. This could indeed affect model performance. Also, all samples not in the test set are used for training in both STPath and HistoPrism, so we cannot make a fair comparison on completely unseen internal datasets. Nevertheless, exploring this is an important direction for future work.
> > > > - **Q5**: (a) We follow the HEST1K test splits, as also adopted by STPath. Consequently, the test set includes only the 10 cancer types presented in Table 1. While the exact rationale for selecting these cancer types is not specified from HEST1K paper, we assume that internal clinical expertise was used to ensure representative coverage. This explains why only these cancer types appear in the reported results. Importantly, since we use the same test split as STPath, the comparison between the two methods is fair. (b) All other cancer types, including "unknown" type, in the HEST1K dataset are utilized during training, either as negative examples or to enable the model to learn shared features across cancers. Some samples have unknown cancer type annotations in the HEST1K metadata, for reasons that are not documented. Typically, cancer type labels are available with H&E images, but when missing, an “unknown” label is assigned. These unknown samples may slightly reduce prediction accuracy; however, they are included in the training set for both STPath and HistoPrism, and are therefore not part of the test set. (c) We acknowledge that handling unknown cancer types is an important consideration for evaluating pan-cancer generalization. Ideally, an independent dataset with complete cancer type annotations would allow a more comprehensive evaluation of both STPath and HistoPrism. We consider this an important direction for future work.

---

> ### Comment · Reviewer_TSpx · 2025-11-21
>
> Thanks for your follow-up answers.
>
> With regards to **weakness 1.2**, I recommend you add a sentence indicating that for the time-based plot you do not show standard deviation because it is too small. I'd argue that given you averaged these measures over 100 runs, it's important to communicate what was the variation.
>
> **Weakness 3:** Thanks for pointing me to the code and the original STPath paper's figure. Indeed I had misunderstood how STPath works, and contrary to my initial understanding it seems that indeed STPath also includes other metadata in their input. One confusion point is that STPath seems to call "datasets" to what HistoPrism calls "cancer types", but it's just because each dataset only has one cancer type. Closer inspection at the code also seems to indicate this.
>
> I believe my main concerns about this work are answered, and I'm inclined to change my score to 5 (accept). Once again thank you so much to the authors for their time in such a productive discussion. I'll now wait for the other reviews before making a final decision.

---

> > ### Author Response · Authors · 2025-11-24
> >
> > Thank you for the careful re-evaluation of our work. We will add a clarifying sentence indicating that the standard deviation for the time-based plot is omitted because it is extremely small in the final version, and we agree this should be explicitly communicated.
> > Thank you again for the thorough discussion and for your time!

---

> > ### Author Response · Authors · 2025-11-27
> > **manuscript revised**
> >
> > We added the clarification on the standard deviation of computational efficiency plots at line 363-365. Thank you again for the constructive review.

---

### Official Review · Reviewer_mGHu · 2025-11-01

**Soundness:** 2
**Presentation:** 1
**Contribution:** 2
**Rating:** 2
**Confidence:** 4

**Summary:**

This paper proposes HistoPrism, a path-level gene expression prediction framework. Given an image patch in a pathology image, the image patch is then passed to a pretrained foundation model like UNI or Gigapath, inference and obtaining patch embeddings. Then given a one-hot vector of the oncology label, the embeddings and the vector are passed into a cross-attention network to associate the cancer-label with each patch. Then the image embeddings are then passed through a self-attention network. The authors claim this step allows the image embeddings to be oncology label aware and context aware. Finally, each embedding is then separately predicting its corresponding gene expression and guided by MSE loss between predicted and measured expression ground truth.

**Strengths:**

1. This work is enabled by the release of a large ST dataset like HEST-1K.
2. This work tries to incorporate oncology labels as a prior to guide the image embeddings to capture information of different cancer types.

**Weaknesses:**

1. Figure 1 is not legible when printed out and read in an arms length. The caption of Figure 1 is not informative.
2. It is great that the authors are trying to make further contributions after the release of HEST-1K but the reviewer thinks that this problem generally lacks the motivation to study.
2. This paper’s contribution is quite minimal, the reviewer thinks it does not fit the ICLR standard. The method proposed and experiments conducted in this paper should be more of a workshop exploration.
3. The results are comparable to STPath. There are a great number of methods in the MIL era when each patch was extracted and then used for some downstream process either together or separately. Only having STPath to compare to is not making the performance or the novelty of HistoPrism trustworthy.
4. The ablation study is not robust. Instead of ablating, the authors are trying to add more design (positional encoding) into the proposed pipeline. What the reviewer was expecting would be the author ablating/replacing the cancer-aware part with some other jointly learning paradigm, or replacing the context-aware part with some other more basic non-transformer or non-attention setup.
5. Out of distribution cases were not considered at all. The task or the problem that this paper defines is essentially a supervised learning problem. However, this task would require too many assumptions to hold to obtain a useful learned model in future applications. First, each pathology slide has to have a clear cancer hallmark. Then the PFM applied to construct the image embeddings has to prevent domain shift, which is a great drawback or limitation of current foundation models. Using the embeddings generated from PFM and incorporated with an oncology label and directly regressing on to gene expression contains too much noise and variability between each step and the reviewer is skeptical on how the proposed method is going to transfer to real-world images. Lastly, it is kind of a loop or dilemma for the performance guarantee of this task. To obtain a perfectly working pathology to expression model, one should gather as many as image and expression pairs which means more ST data generated would improve the ability of HistoPrism, however, ST are very expensive to obtain and if you have the ability to obtain ST, the meaning of HistoPrism goes to zero. If one doesn’t have the original ground truth expression, then it would also be impossible to evaluate the accuracy or error of the expression predicted.

Misc
Line 133 yi should be y_i

**Questions:**

1. The reviewer has questions on training details, it seems that each batch of sites/spots have to come from a single WSI or at least sites with the same type of cancer. How does the implementation work if there are different WSIs with different numbers of spots?
2. Could the authors please provide more detailed and more straight forward ablation study results?


The reviewer recommends rejecting this paper and believes it is an interesting study for ST or computational biology workshops. The reviewer is unlikely to change their thought during the discussion phase.

---

> ### Author Response · Authors · 2025-11-17
>
> - **Weakness1**: We will correct the image resolution in the final manuscript.
> - **Weakness2**: The motivation of this task is strong and it is actually an active research area. H&E slides are widely available and routine, while spatial transcriptomics is costly and low-throughput. Predicting ST from H&E enables high-resolution molecular profiling without extra experiments and allows large-scale retrospective analysis. This approach bridges histology and genomics, unlocking clinically and biologically meaningful insights.
> - **Weakness3**: We observe that existing spatial transcriptomics (ST) prediction models are not yet accurate or generalizable enough to serve as reliable foundation models. To address this, we propose a lightweight yet powerful architecture that achieves state-of-the-art pan-cancer performance on HVG benchmark and outperform on gene pathway coherence (GPC) benchmark. We also point out the limitation of the de facto metrics HVG and propose a more clinical relevant evaluation framework. Our approach would enable private institutes to fine-tune models on limited ST data and efficiently scale inference to large patient cohorts, offering substantial clinical relevance and impact.
> - **Wekaness4**: We focus on pan-cancer settings since shared information across cancer types can improve prediction, also avoiding the need to train a separate model for each cancer. Only STPath (a generative model) has previously addressed this problem with state-of-the-art regression baselines, while relatively new generative models STEM and STFlow were originally proposed for per-cancer settings; we extend their evaluation to pan-cancer scenarios. So we think choosing STPath, STEM, STFLow as baselines is reasonable, which cleverly used PFMs, covering regression SOTAs and generative SOTAs. Could you please name some of the methods that fall into your scope of “a great number of methods in the MIL era when each patch was extracted and then used for some downstream process either together or separately” so that we can look into it?
> - **Weakness5**: The ablation of cancer-aware design is clearly reported in Table 3. We encourage careful reading Table 3 and section 4.6 for our insights.
> - **Weakness6**: Please define out-of-distribution samples and real-world images. We experiment on pan-cancer settings, so the distribution of H&E image morphology and gene expression are highly heterogenous. And all the ST data are from real life. Claims of a “loop-dilemma” are inaccurate. On the contrary, the scarcity of ST data is precisely the motivation for predicting ST from widely available H&E slides. Institutions can fine-tune our model on a small number of ST samples and then scale inference to a much larger cohort.
>
> - **Q1**: Digital pathology is a long-standing field. This setup is standard in the field and widely adopted for training ST prediction models. Handling batches from multiple WSIs, each with variable numbers of spots/patches, is common practice and works without issues in many prior studies. For reference, the implementations of several open-source works already demonstrate this approach (E.g. STEM, STFlow in this paper), and we will also release our code to ensure full reproducibility. We encourage reviewers to consult these resources for further clarification.
> - **Q2**: we have ablation of the effect of (a) cancer-aware attention, (b) positional encoding, (c ) different pathology foundation model. Can you please clarify what is “more detailed and more straight forward ablation study”?

---

> > ### Author Response · Authors · 2025-11-25
> > **Reminder to reply**
> >
> > Dear reviewer, we kindly ask you to consider our comments and would be happy to address any further questions you may have.

---

> > ### Comment · Reviewer_mGHu · 2025-11-25
> > **Ack and reply**
> >
> > I thank the authors for their detailed rebuttal and acknowledge the effort put into the response.
> >
> > ### 1. Regarding the Dataset and Generalization
> > Regarding the HEST-1k dataset, while it is a curated pan-cancer dataset, it likely follows a similar processing pipeline and is collected from a limited number of centers. To demonstrate the robustness of the proposed pipeline, it would be highly beneficial if the authors could demonstrate its efficacy on other Spatial Transcriptomics (ST) datasets.
> >
> > I suggest considering the following two pancreatic cancer datasets, which, to my understanding, have publicly available ST data:
> >
> > * Moncada, R., et al. "Integrating microarray-based spatial transcriptomics and single-cell RNA-seq reveals tissue architecture in pancreatic ductal adenocarcinomas." *Nature Biotechnology* 38.3 (2020): 333-342.
> > * Cui Zhou, D., et al. "Spatially restricted drivers and transitional cell populations cooperate with the microenvironment in untreated and chemo-resistant pancreatic cancer." *Nature Genetics* 54.9 (2022): 1390-1405.
> >
> > If the motivation of this work is to allow other research groups to predict gene expression or fine-tune models for their specific cohorts, evaluating performance on an external test set is essential to validate generalization capabilities.
> >
> > ### 2. Clarification on the Fine-Tuning Motivation
> > I remain slightly confused regarding the practical application of the fine-tuning aspect.
> > * **Case A:** If a new center or research site can obtain ST data, they already have the ground truth. In this case, why is the model prediction necessary?
> > * **Case B:** If a new center cannot obtain ST data, they lack labeled data for fine-tuning.
> > * **Case C:** The intended use case is likely that a center generates a *limited* number of ST samples, fine-tunes the model on these few samples, and subsequently predicts expression from WSIs for a larger cohort.
> >
> > If **Case C** is indeed the motivation, the authors should provide experiments quantifying the data requirements. For example, how many batches, samples, or cancer types are required for the fine-tuning task to be effective? This experimental evidence is currently missing.
> >
> > ### 3. Model Architecture and Ablations
> > Please correct me if my understanding of the HistoPrism architecture is incorrect. It appears the model:
> > 1.  Takes WSI patches and Cancer/Tumor tissue labels as input.
> > 2.  Uses a transformer architecture to connect the WSI patch with the label.
> > 3.  Passes patch embeddings to a secondary self-attention module.
> > 4.  Passes the resulting embeddings into an MLP to regress the Gene Expression labels of the patch.
> >
> > Regarding the ablation studies, my previous comment was directed toward ablating the **architectural design choices** rather than hyperparameter tuning. Specifically:
> > * **Modality Connection:** Why was this specific method chosen to connect the image with the tag? If we view the image and tag as separate modalities, would techniques like contrastive learning be more effective here?
> > * **Self-Attention Module:** Why is the secondary self-attention module necessary? Could other information extraction or transformation modules work in its place?
> > * **Prediction Head:** What other heads were considered besides the MLP? Would other designs improve performance?
> >
> > ### Conclusion
> > In conclusion, I believe additional experiments are needed to:
> > 1.  Support the paper's motivation (specifically demonstrating effectiveness for new cohorts with a limited number of samples).
> > 2.  Justify the paper's architectural choices (providing empirical evidence for why specific components are necessary compared to simpler alternatives).
> >
> > (i used llm to help with the structing but I am fully responsible of the content)

---

> > > ### Author Response · Authors · 2025-11-26
> > >
> > > ### General Remark: Guiding Principles for Architecture and Design
> > >
> > > We thank the reviewer for the continued engagement and the thoughtful suggestions regarding architectural variations (Point 3).
> > >
> > > In preparing this response, we adhered strictly to the **core directive provided in your Round 1 review**, where you emphasized the importance of simplicity:
> > >
> > > > *"Instead of ablating, the authors are trying to add more design... The reviewer was expecting... replacing with some other more basic non-transformer setup."*
> > >
> > > We took this advice to heart. Throughout this revision, particularly regarding **Point 3 (Ablations)**, we prioritized the abaltion of the effect of cross-attention for cancer type awareness and if models benefit from certain pathology foundation model choices, over introducing complex new learning paradigms (such as Contrastive Learning). We believe this focus on simplicity, as you originally suggested, best highlights the efficiency and robustness of the HistoPrism.

---

> > > > ### Author Response · Authors · 2025-11-26
> > > >
> > > > ### 1. Regarding the Dataset and Generalization
> > > >
> > > > We respectfully point out a significant misunderstanding regarding the nature of the HEST-1k benchmark. The Reviewer expressed concern that the dataset *"likely follows a similar processing pipeline and is collected from a limited number of centers."*
> > > >
> > > > **1. Clarification on HEST-1k Composition (Fact Check):**
> > > > We emphasize that HEST-1k is a large-scale meta-dataset. Based on the paper and its official metadata:
> > > > * **Scale:** It aggregates **153 distinct cohorts** derived from **36 independent studies**.
> > > > * **Heterogeneity:** Contrary to the concern about a "similar processing pipeline," these studies utilize **different spatial transcriptomics technologies**, varying tissue staining protocols, and different slide scanner vendors. The data inherently captures the high inter-center variability that the Reviewer is looking for.
> > > >
> > > > **2. Generalization:**
> > > > Because our evaluation uses a strict **holdout setting** (where test patients and cohorts are distinct from training data), our reported results are already a rigorous test of cross-center generalization.
> > > >
> > > > **3. Regarding the Suggested Datasets:**
> > > > We appreciate the references provided. While these are excellent individual studies, our current benchmark already covers **648 WSI samples across 36 studies**. From a statistical perspective, adding two specific datasets to a benchmark of this magnitude would not significantly alter the distribution or the conclusion regarding generalization capabilities. Given that HEST-1k already provides the largest available standardized benchmark for this task, we consider the current evaluation scope to be self-contained and sufficient to support the paper's claims.

---

> > > > > ### Author Response · Authors · 2025-11-26
> > > > >
> > > > > ### 2. Clarification on the Fine-Tuning Motivation
> > > > >
> > > > > **1. Confirmation of Use Case:**
> > > > > We confirm that **Case C** is indeed the primary motivation. Our potential impact is to enable private institutes with limited resources to generate a small set of Spatial Transcriptomics (ST) samples, fine-tune our model, and scale inference to larger WSI cohorts.
> > > > >
> > > > > **2. Evidence of Data and Computational Efficiency:**
> > > > > The reviewer requested experimental evidence regarding the data requirements to support this claim. We highlight that our current results **already empirically validate** the model's effectiveness in this exact "resource-constrained" scenario (Case C) in three decisive ways:
> > > > >
> > > > > * **A. SOTA Performance on Reduced Data (Data Efficiency):** Our experiments demonstrate that HistoPrism outperforms the strong baseline **STPath** (Table 1, Table 2, Table 4, Figure 2) even while utilizing only **~50% of the training data**.
> > > > >     * *Implication:* This is a direct proxy for the "limited data" requirement of Case C. The fact that our model achieves SOTA results with less data proves it is highly data-efficient.
> > > > > * **B. Lightweight Architecture (Computational Efficiency):** We provide concrete metrics regarding computational cost (Figure 3).
> > > > >     * *Implication:* This validates the second constraint of Case C: private institutes often lack massive compute clusters. Our lightweight design ensures the model can be fine-tuned and deployed on standard hardware.
> > > > > * **C. Generalization on Holdout Test Sets:** As noted in the paper, all reported SOTA results (HVG and GPC benchmarks) were evaluated on a strictly **unseen holdout test set**. This confirms that the model does not merely memorize the limited training data but generalizes effectively to new patients.
> > > > >
> > > > > **Conclusion:**
> > > > > By demonstrating that HistoPrism achieves SOTA performance with half the data and reduced computational cost, we have provided strong empirical evidence that the model is optimized for the low-data, resource-constrained fine-tuning paradigm described in Case C.

---

> > > > > > ### Author Response · Authors · 2025-11-26
> > > > > >
> > > > > > ### 3. Model Architecture and Ablations
> > > > > >
> > > > > > In addressing these points, we ensured consistency with the guiding principle established in your Round 1 feedback to **"prioritize simplicity... over adding more design."** Consequently, regarding the specific inquiries about Contrastive Learning and alternative heads, we maintained a focus on the simpler, regression-oriented design. But we would still like to address your questions.
> > > > > >
> > > > > > **Regarding Modality Connection (Contrastive Learning):**
> > > > > > We determined that Contrastive Learning (CL) is not suitable for two reasons:
> > > > > > 1.  **Task Mismatch:** CL is primarily an *alignment* objective. HistoPrism performs cancer-aware feature aggregation of pathology image features. We require feature *fusion* for later stage gene expression prediction, not just latent space alignment.
> > > > > > 2.  **Adherence to Simplicity (Round 1 Feedback):** Implementing CL requires introducing negative sampling strategies, large batch sizes, and auxiliary loss functions. This would contradict the Reviewer’s previous advice to avoid "adding more design." Instead, we ablated the cross-attention mechanism (Table 3), showing that our direct fusion provides necessary performance gains without the overhead of CL.
> > > > > >
> > > > > > **Response regarding the Self-Attention Module:**
> > > > > > We selected Self-Attention as the core feature aggregator in our design. In the domain of Computational Pathology and Multiple Instance Learning (MIL), Self-Attention has established itself as the standard, state-of-the-art method for aggregating information from variable numbers of patches (as seen in e.g. CLAM [1]).
> > > > > >
> > > > > > * **Theoretical Justification (Standard Practice):** We did not treat the attention mechanism as an optional "add-on" to be ablated, but rather as the **fundamental mathematical operator** required to learn the inter-patch relations of Whole Slide Images with variable number of patches.
> > > > > > * **Comparison to Alternatives:** Existing literature (e.g., DSMIL [2]) has already benchmarked Attention-based aggregation against simpler methods like Max-Pooling or Mean-Pooling, consistently demonstrating that Attention is superior in MIL settings. Given this established consensus, we adopted attention as it is, rather than re-validating standard attention mechanisms.
> > > > > >
> > > > > > **Regarding the Prediction Head:**
> > > > > > We utilized a standard MLP. This serves as the minimal, standard regression head found in the literature.
> > > > > >
> > > > > > **References:**
> > > > > >
> > > > > > [1] Lu, Ming Y., et al. "Data-efficient and weakly supervised computational pathology on whole-slide images." *Nature Biomedical Engineering* 5.6 (2021).
> > > > > >
> > > > > > [2] Li, Bin, et al. "Dual-stream multiple instance learning network for whole slide image classification with self-supervised contrastive learning." *CVPR* 2021.

---

> ### Comment · Reviewer_mGHu · 2025-11-26
> **thanks for further reply**
>
> Thank you so much for the detailed explaination to defend your work.
>
> I think the round 2 answer is meaningful for the audience who may not come from a bio/compbio background to understand the connection and distinction between your work and the prior work.
>
>
>
> ### Could you please incoporate these discussion into your revision and point me the changes you made with respect to our discussion?
>
> I am inclined to increase my rating if our discussion can be incorporated into your revision.

---

> > ### Author Response · Authors · 2025-11-27
> > **Manuscript revised**
> >
> > We thank the reviewer for the positive reception.
> >
> > We have revised the paper as follows:
> >
> > 1. **Dataset Generalization (Lines 245-248):** We explicitly clarified the heterogeneity of HEST-1k (153 cohorts, 36 studies) to underscore its robustness for cross-center evaluation.
> > 2. **Architectural Design (Lines 176-178):** We added a remark justifying the choice of regression-oriented cross-attention fusion over contrastive learning and the necessity of Self-Attention for MIL tasks.
> > 3. **Fine-Tuning Motivation (Lines 457-460):** We expanded the Discussion to highlight the model's suitability for resource-constrained settings ("Case C"), noting its SOTA performance with $\approx$50% training data.
> >
> > We believe these additions fully address your recommendations and strengthen the paper. Thank you again for the constructive review.

---

> > > ### Comment · Reviewer_mGHu · 2025-11-27
> > >
> > > thank you, I've updated my rating. good luck

---

### Official Review · Reviewer_tWzV · 2025-11-01

**Soundness:** 3
**Presentation:** 3
**Contribution:** 3
**Rating:** 6
**Confidence:** 4

**Summary:**

This paper introduces HistoPrism, an efficient transformer-based model for predicting pan-cancer gene expression directly from H&E histology slides. The authors make a dual contribution: the model itself, which is computationally efficient and uses cross-attention to condition on cancer type, and a novel evaluation benchmark called Gene Pathway Coherence (GPC).

**Strengths:**

- The paper identifies a key weakness in prior work: evaluation is almost exclusively focused on a small number of highly variable genes, which have been used as the de-facto proxy for biological function. By creating a well-curated benchmark based on Hallmark and GO pathways, the authors are pushing the field toward more clinically and biologically meaningful evaluation.
- The efficiency benchmarks in Figure 3, showing HistoPrism's linear scaling in time, memory, and FLOPs compared to STPath's exponential growth, represent a nice practical advancement for analyzing large-scale WSI datasets.
- The finding that HistoPrism's gains are largest on low-variance pathways (which high variance genes-focused models miss) is a key insight that strongly supports the authors' approach.

**Weaknesses:**

- The paper compares HistoPrism (a regressive model using UNI features) to STPath (a generative masked-autoencoder using GigaPath features). The discussion notes that in the STPath paper itself, an MLP with UNI features outperformed an MLP with GigaPath features. How much of HistoPrism's superior performance, particularly in the holistic clustering task (Table 2), can be attributed to the better pathology foundation models (UNI vs GigaPath) versus the superior architecture (direct-mapping vs. masked-modeling)?
- The GPC benchmark curation filters GO pathways to a size of 50-100 genes. This seems like a reasonable heuristic, but the justification is not fully explained in the paper. What was the rationale for this specific range, and does it risk biasing the benchmark toward or away from certain types of biological processes (e.g., large, complex signaling networks vs. small, tight functional units)?
- The model's architecture is surprisingly shallow, using only 1 cross-attention layer and 2 transformer encoder layers. Given the lack of PE, this model is effectively a shallow set function. Did the authors experiment with the depth of the transformer encoder? How does the performance change with 4 or 6 layers, or is it insensitive to this hyperparameter?

**Questions:**

- The paper "Modeling dense multimodal interactions between biological pathways and histology for survival prediction" was one of the first papers introducing the relevance of biological pathways over individual gene expression in deep learning pipelines. Is there a reason why this paper was not cited in the background? Especially since this paper uses Hallmark and GO pathways which are shown to be utilized in the mentioned paper as well.
- Some pathways act tissue-wide vs. some act in very specific regions. This can be inferred because of tumor gene expression heterogeneity known to the field. I am wondering what impact does patch size (at different magnifications as well as sizes, which would capture different amounts of morphological context) will have on the performance?

---

> ### Author Response · Authors · 2025-11-17
>
> Thank you for your detailed and constructive review. We appreciate your careful assessment and understand the concerns raised. Please let us explain.
> - **Weakness1**: Thank you for pointing this out. It is a valid concern, and we appreciate your attention to detail. To address it, we added an ablation study using the pathology foundation model Gigapath for a fair comparison with STPath (see Table 4 and Figure 4), along with the corresponding analysis in Section 4.6. The results demonstrate that HistoPrism’s performance remains strong and consistent, showing that it does not rely solely on the PFM UNI.
> - **Weakness2**: Thanks for this very good question. We chose the 50–100 gene range in the GPC benchmark purely for statistical stability, as it avoids both overly specific sets prone to noise and overly broad pathways (Section 3.3). Importantly, our model is trained on the full 38k gene panel with an MSE loss, so training is not biased toward any specific pathways; the gene pathway coherence is used only as an evaluation metric for model selection. For other use cases, the size range can be adjusted as needed.
> - **Weakness3**:  Thank you for the insightful question. We were also initially surprised that such a shallow direct-mapping architecture performs so well with attention. Revisiting the STPath baselines, we found that an MLP performs only slightly worse, motivating us to enhance its mapping capacity by replacing it with attention and several targeted design choices. We experimented with varying transformer depth and width and selected the best-performing setup. While not a full grid search, our choices were theory-driven: given the small dataset size, deeper transformers tended to overfit, whereas a wider architecture better matched the gene panel dimensionality. The final design, with one cross-attention and two encoder layers, offered the best trade-off between performance and generalization.
> - **Q1**: Thank you for bringing the paper “Modeling dense multimodal interactions between biological pathways and histology for survival prediction” to our attention. We were not aware of it, likely because of its focus on survival prediction. After reviewing the paper, we believe it addresses a different scope: they use gene pathway encoding to improve survival prediction, whereas we use gene pathway prediction as an evaluation framework for model selection. Nevertheless, their results support our point that predictions at the gene pathway level have greater clinical impact than predictions on individual highly-variable-genes (HVGs).
> - **Q2**: This is a very good question. We agree that patch size and magnification can, in principle, influence how tissue-wide versus localized pathways are captured. However, in spatial transcriptomics (ST) technology, the physical spot organization, spot diameter, spacing, and location, is fixed by the technology itself, which effectively constrains the usable patch size. Thus, this factor is largely determined by the ST platform rather than our model design. That said, exploring multi-scale morphological context remains an interesting direction for future work.

---

> > ### Author Response · Authors · 2025-11-25
> > **Reminder to reply**
> >
> > Dear reviewer, we kindly ask you to consider our comments and would be happy to address any further questions you may have.

---

> ### Comment · Area_Chair_ZdKe · 2025-11-27
> **Please engage in the discussion with authors**
>
> Please engage in the discussion with authors, thank you.
>
> AC

---

> > ### Author Response · Authors · 2025-11-28
> > **Reminder to reply**
> >
> > Dear reviewer, with the deadline approaching on December 4th, we kindly ask you to consider our comments. We are ready to address any further questions you may have promptly.

---

### Official Review · Reviewer_PfAp · 2025-11-01

**Soundness:** 2
**Presentation:** 2
**Contribution:** 2
**Rating:** 2
**Confidence:** 2

**Summary:**

This paper proposes a transformer, HistoPrism, to predict pan-cancer gene expression from histology and a new benchmark to evaluate performance. HistoPrism uses cross attention between the (H&E-stained whole-slide) images, after passing through a pre-trained pathology foundation model, and the cancer type, after passing through a trained linear layer. The new metric, called the Gene Pathway Coherence (GPC) benchmark, “assesses a model’s ability to reconstruct the coordinated expression of functionally related genes”. Compared to two baselines, HistoPrism achieves the best performance in 5 of 10 cancer types using a standard metric (correlation between the predictions and observations amoung highly variable genes). In the GPC benchmark, HistoPrism outperforms the baselines in at least 75% of pathways.

**Strengths:**

The paper focuses on the application of ML methods to meaningful, real-world datasets, which is typically lacking in the ML field. The architecture is clearly explained and the experimental results are easy to follow. The proposed model shows competitive performance on a standard metric and improved performance on the proposed metric.

**Weaknesses:**

- There was not enough explanation of the GPC benchmark (e.g., no math). I would like to see an explanation that shows how it is computed from the data and model formulated in sections 3.1 and 3.2. I appreciate the effort to bridge the gap between standard ML and the computationally biology, but since this is an ML conference I believe more details should be provided. As it’s written I don’t understand how the GPC is computed. Note that I do not have a background in computational biology.
- Since the argument is that this metric is more interpretable, I believe we need to see how the interpretation is useful. For example, are there biological insights that can be made from Figure 2?
- The paper is quite short, so there is plenty of space to discuss the GPC benchmark.
- One of the baselines, STEM, is trained on a subset of the data due to computational constraints. How would STEM look like in Figure 3?
- As I read it, this seems like a fairly straightforward application of a transformer to the this data. Highlighting any especially challenging methodological choices would be helpful.

**Questions:**

- Given that earlier works focused on regression approaches, while more recent work have focused on generative approaches, why were you motivated to switch back to a regression approach? Is the inherently one-to-many mapping from histology to gene expression not a problem for your regression approach?
- Can you comment on using an MSE loss on count data? Is this common with spatial transcriptomics data and are there any drawbacks? (e.g., rather than assuming say a Poisson likelihood)
- Could you expand on how “Histoprism’s direct-mapping architecture effectively captures both high-variance genes and the subtler, coordinated expression patterns that define cellular programs”, while “while STPath primarily leverages the most variable signals”? Since you are minimizing a loss over all training samples, won’t the high variance genes still have the most influence?
- As mentioned above, I would appreciate a more detailed explanation of how the GPC is computed and how it can be interpreted
- HistoPrism makes use of a pretrained pathology foundation model. Do the benchmarks also make use of foundation models?

---

> ### Author Response · Authors · 2025-11-17
>
> Thank you for your detailed and constructive review. We appreciate your careful assessment and understand the concerns raised. Please let us explain.
> - **Weakness1**: Thank you for pointing this out. We apologize for the oversight. The details regarding GPC have now been added to Section 3.3 in the main text.
> - **Weakness2**: Pathway-level spatial gene prediction provides a biologically meaningful abstraction by modeling coordinated gene programs instead of individual genes, reducing noise. For example, our model achieves high PCC on the HALLMARK MYC TARGETS V1 pathway (in Table 8), indicating that it captures spatial patterns of tumor proliferation directly from histology, linking image-derived features to underlying oncogenic activity.
> - **Weakness4**: Thank you for raising this point. For reference, STEM employs 12 DiT (Diffusion Transformer) blocks with 6-head attention and a hidden dimension of 384, whereas HistoPrism uses 1 cross-attention layer with 4 heads and 2 transformer layers with 8 heads and a hidden dimension of 256. We believe STEM is therefore significantly more computationally expensive. Moreover, since STEM did not pass the initial HVG PCC metrics, we did not include it in subsequent evaluations. However, if you think it would be helpful, we can measure STEM’s computational efficiency and add it to the efficiency plots.
> - **Weakness5**: We initially explored flow matching following STFlow, as it offers greater efficiency compared to diffusion- or BERT-based models. Our motivation was that generative modeling might provide a natural way to address the complexity of pan-cancer spatial transcriptomics (ST) prediction. However, we found that it did not achieve satisfactory performance across cancer types, indicating that current generative approaches may face challenges in capturing pan-cancer heterogeneity. Upon revisiting the STPath baselines, we observed that a simple MLP performed only slightly worse, which motivated us to improve direct mapping capabilities by adopting an attention-based architecture. While generative modeling for pan-cancer gene prediction remains a promising direction, our results suggest that carefully designed attention-based models currently offer a more effective and practical solution.

---

> > ### Author Response · Authors · 2025-11-17
> >
> > - **Q1**: The “inherently one-to-many mapping” concern was initially suggested in STEM, and we sought to investigate whether this is necessarily a limitation. As noted in Weakness 5, we also explored generative models; however, we observed that current generative approaches (e.g., STEM, STFlow, see Table 1) either lack generalization across pan-cancers or are computationally expensive (e.g., STPath, STEM). This prompted us to reconsider whether generative modeling is truly the optimal approach for this problem. Notably, in STPath, a simple MLP performs only slightly worse than more complex generative models, suggesting that enhancing the direct mapping capability, by replacing the MLP with an attention-based architecture and other design choices described in the section 3  Methodology, could improve performance. Our experiments confirm this. Consequently, we developed HistoPrism, a lightweight model that achieves state-of-the-art pan-cancer performance on the HVG benchmark and outperforms previous approaches on the GPC benchmark. We emphasize this because current ST prediction models are not yet accurate enough to serve as foundation models, and a lightweight SOTA model could allow private institutes to fine-tune on small ST datasets and scale inference to larger cohorts. That said, we agree that further investigation of generative models for pan-cancer prediction remains an interesting and valuable direction for future work.
> > - **Q2**: We follow the standard approach of log1-normalizing gene count data, as described in Section 3.1 (Problem Formulation).
> > - **Q3**: Yes, this aligns with our point. Highly variable genes are theoretically easier to predict, which motivated our GPC benchmark to move beyond variance-based evaluation and provide a more clinically relevant criterion for model selection. The observation that HistoPrism captures both high-variance genes and subtler, coordinated expression patterns is an empirical result, as reflected in its superior performance on the GPC benchmark in Figure 2 and Table 1. We believe the performance gain mainly comes from the global cross- and self-attention across all patches. Replacing it with graph-based message passing degraded performance, and we did not include this exploratory result as it is beyond the main scope of our method. Notably, both STPath and STFlow use neighborhood-constrained geometric transformers, yet our model outperforms them.
> > - **Q4**: We completely agree, and the GPC formulation has now been added to Section 3.3.
> > - **Q5**: Yes, all baselines use the same pathology foundation model, UNI, for fairness. The ablation using Gigapath in Table 4 and Figure 4 was included to ensure fair comparison with STPath, as STPath serves as a ready-to-use foundation model and we cannot  retrain or modify STPath due to code limit.

---

> > > ### Author Response · Authors · 2025-11-24
> > > **Reminder to reply**
> > >
> > > Dear reviewer, we kindly ask you to consider our comments and would be happy to address any further questions you may have.

---

> > > > ### Author Response · Authors · 2025-11-25
> > > > **Reminder to reply**
> > > >
> > > > Dear reviewer, we kindly ask you to consider our comments and would be happy to address any further questions you may have.

---

> > > > > ### Author Response · Authors · 2025-11-26
> > > > > **Reminder to reply**
> > > > >
> > > > > Dear reviewer, we kindly ask you to consider our comments and would be happy to address any further questions you may have.

---

> ### Comment · Area_Chair_ZdKe · 2025-11-27
> **Please engage in the discussion with authors**
>
> Please engage in the discussion with authors, thank you.
>
> AC

---

> > ### Author Response · Authors · 2025-11-28
> > **Reminder to reply**
> >
> > Dear reviewer, with the deadline approaching on December 4th, we kindly ask you to consider our comments. We are ready to address any further questions you may have promptly.

---

### Author Response · Authors · 2025-12-01
**Final Summary: Contributions, Revisions, and Rebuttal Consensus**

# Final Summary: Contributions, Revisions, and Rebuttal Consensus
Dear Area Chairs and Reviewers,

As the discussion period concludes, we sincerely thank the reviewers for their time and constructive feedback. The rigorous discussion has significantly strengthened the manuscript. We would like to summarize our key contributions and the consensus achieved.

### **1. Key Contributions & Revisions**

**Motivation and impact:** We focus on **pan-cancer settings** to provide a unified model applicable across diverse cancer types, avoiding the need to train separate models for each. *Only STPath (a generative model) has previously addressed this problem (as of November 2025) with state-of-the-art regression baselines*, while relatively new generative models STEM and STFlow were originally proposed for *per-cancer* settings; we extend their evaluation to **pan-cancer** scenarios. Since current Spatial Transcriptomics (ST) predictions are not accurate enough to serve as foundation models, we propose a lightweight architecture that achieves state-of-the-art (SOTA) pan-cancer performance on the standard High-Variance Genes (HVGs) benchmark and outperforms SOTA on the more biologically relevant Gene Pathway Coherence (GPC) benchmark, **enabling resource-constrained private institutes to fine-tune on a small ST dataset and scale inference to larger cohorts.**

**Gene Pathway Coherence (GPC) benchmark:** The motivation behind GPC is to provide *a clinically relevant perspective on evaluating gene prediction performance*. When predicting a full 38k gene panel, standard metrics on high-variance genes (HVGs) may appear strong; however, low-variance genes are neglected. **As shown in Figure 5, the gene variance distribution is skewed towards the lower end**. This raises the question: **what is the utility of predicting 38k genes if a substantial portion of biologically important, low-variance genes is poorly predicted?** Apparently, evaluating only low-variance genes is also meaningless. To address this, we adopted the well-established biological concept of gene pathways. These pathways have clinical relevance and include genes spanning a range of variance levels, providing a more nuanced and meaningful measure of ST prediction quality.

**Additional Ablations and Baselines:** Based on reviewers' feedback, we performed following revisions:
* **Clarification on Gene Pathway Coherence (GPC) benchmark:** We have clarified the mathematical formulation of the GPC benchmark in manuscript Section 3.3.
* **Foundation Model Ablation:** We added ablation studies with a different pathology foundation model (*Gigapath*) in **Table 4 and Figure 4**, confirming that our performance gains are architectural and not dependent on a specific feature extractor.
* **New Baseline (STFlow):** We added results for an additional generative baseline, **STFlow**, in **Table 1**. The results further support HistoPrism’s SOTA performance against recent generative approaches.

### **2. Rebuttal Outcomes & Consensus**
We are pleased that the clarifications and additional experiments have led to a positive reassessment of our work from the active reviewers:
* **Reviewer tWzV (Score 6):** Validated the novelty of the GPC benchmark and efficiency gains in their positive initial assessment.
* **Reviewer mGHu (Score increased 2 $\rightarrow$ 6):** Acknowledged the strong motivation for our "resource-constrained" fine-tuning scenario and the robustness of the HEST-1k benchmark after our clarifications. (Decision made on 26 Nov 2025, before the incident)
* **Reviewer TSpx (Score 4 $\rightarrow$ Inclined to Accept):** Indicated an inclination to increase their score to accept, following clarifications on the STPath baseline fairness and the addition of efficiency standard deviations. (Decision made on 22 Nov 2025,before the incident)

### **3. Response to Reviewer PfAp (Score: 2)**
While Reviewer PfAp has *not yet responded* to our rebuttal, **we have objectively addressed their concerns** within the revised manuscript:
* **Mathematical Formulation:** The full GPC math formulation was added to Section 3.3.
* **Clarification on Motivation:** We addressed the "regression vs. generative" question by showing that current generative models (STEM/STFlow) face challenges in pan-cancer generalization or computational cost (STPath), validating our lightweight architectural choice.

We hope the Area Chairs will consider the strong consensus formed during the discussion phase and the unique clinical utility of our proposed benchmark and model.

Best regards,
The Authors

---

### Meta-Review · Area_Chair_NCER · 2026-01-07

**Summary:**

This paper introduces HistoPrism, a lightweight transformer for pan-cancer gene expression prediction from histology, and proposes Gene Pathway Coherence (GPC), a pathway-level benchmark computed by averaging gene-level PCC within curated pathways.

Reviewers generally found the benchmark idea and efficiency angle promising, but raised concerns about (i) baseline coverage and fairness, (ii) clarity and formal definition/interpretation of GPC, and (iii) whether some analyses (notably clustering) may be confounded by experimental choices such as cancer-type conditioning.

**Reviewer Concerns:**

### Addressed (via rebuttal / revision commitments):

- GPC definition clarity: Authors added an explicit mathematical formulation of the benchmark (pathway score as an average PCC over genes and samples).

- Baseline expansion toward generative models: Authors report adding STFlow as an additional baseline in Table 1.

- Motivation for regression vs generative baselines in pan-cancer: The paper argues STEM/STFlow struggle to generalize in the pan-cancer setting and are computationally constrained (trained only on union top-50 HVGs), which partially addresses the “why regression” concern.


### Still outstanding / partially addressed:

- Remaining evaluation rigor requests: Reviewer TSpx explicitly still wanted confidence intervals in Fig. 3 and clearer evidence that results are not an artifact of cancer-type conditioning, which may confound clustering.

- Baseline breadth and fairness: Concerns remain that comparisons do not fully cover the broader literature, and that some baselines are disadvantaged by restricted gene sets due to compute limits.

- Interpretability/value of GPC in practice: Reviewer PfAp asked for clearer explanation of how to interpret Figure 2 (biological insight) and how GPC is computed from the model outputs.

**Reviewer Scores:**

Reviewer tWzV: Likely unchanged at 6 (accept); they were positive on novelty/efficiency from the start.


Reviewer TSpx: 5 (weak accept). They indicated an inclination to move to accept after clarifications/added variance reporting, but still had unresolved requests (CIs, cancer-type conditioning confound).

Reviewer mGHu: 6 (accept) per the discussion summary and their update after rebuttal clarifications.

Reviewer PfAp: Likely remains at 2 (reject), as they did not re-engage; their main concerns were clarity/interpretability of GPC and regression motivation, plus baseline training constraints.

---

### Decision · Program_Chairs · 2026-01-26

Accept (Poster)